# Structural mechanism of ATP-independent transcription initiation by RNA polymerase I

Yan Han[1], Chunli Yan[2,3‡], Thi Hoang Duong Nguyen[4‡], Ashleigh J Jackobel[5], Ivaylo Ivanov[2,3], Bruce A Knutson[5*], Yuan He[1*]

[1]Department of Molecular Biosciences, Northwestern University, Evanston, United States; [2]Department of Chemistry, Georgia State University, Atlanta, United States; [3]Center for Diagnostics and Therapeutics, Georgia State University, Atlanta, United States; [4]Howard Hughes Medical Institute, University of California, Berkeley, United States; [5]Department of Biochemistry and Molecular Biology, SUNY Upstate Medical University, Syracuse, United States

**Abstract** Transcription initiation by RNA Polymerase I (Pol I) depends on the Core Factor (CF) complex to recognize the upstream promoter and assemble into a Pre-Initiation Complex (PIC). Here, we solve a structure of *Saccharomyces cerevisiae* Pol I-CF-DNA to 3.8 Å resolution using single-particle cryo-electron microscopy. The structure reveals a bipartite architecture of Core Factor and its recognition of the promoter from −27 to −16. Core Factor's intrinsic mobility correlates well with different conformational states of the Pol I cleft, in addition to the stabilization of either Rrn7 N-terminal domain near Pol I wall or the tandem winged helix domain of A49 at a partially overlapping location. Comparison of the three states in this study with the Pol II system suggests that a ratchet motion of the Core Factor-DNA sub-complex at upstream facilitates promoter melting in an ATP-independent manner, distinct from a DNA translocase actively threading the downstream DNA in the Pol II PIC.

*For correspondence: knutsonb@upstate.edu (BAK); yuanhe@northwestern.edu (YHe)

‡These authors also contributed equally to this work

Competing interests: The authors declare that no competing interests exist.

## Introduction

Eukaryotic RNA synthesis is catalyzed by at least three classes of RNA Polymerases (Pol I-III) (*Roeder and Rutter, 1969*). The large ribosomal RNA precursor (pre-rRNA) is transcribed by Pol I (*Moss et al., 2007*), accounting for up to 60% of total cellular RNA synthesis in *Saccharomyces cerevisiae* (*Warner, 1999*). Transcription by Pol I is highly regulated, and its mis-regulation has been implicated in many diseases including various types of cancer (*Drygin et al., 2010*; *Montanaro et al., 2013*; *White, 2008*).

Pre-Initiation Complex (PIC) formation is a key regulatory step in the control of gene transcription by eukaryotic RNA polymerases. Yeast Pol I transcription initiation is regulated by four general transcription factors: the regulatory factor Rrn3, the Core Factor (CF), the TATA-box Binding Protein (TBP), and the Upstream Activation Factor (UAF) (*Schneider, 2012*). Rrn3 contains an elongated HEAT repeat (*Blattner et al., 2011*), and binds Pol I via contacts with subunits A43, A190, and AC40 (*Blattner et al., 2011*; *Cavanaugh et al., 2008*; *Engel et al., 2016*; *Milkereit and Tschochner, 1998*; *Peyroche et al., 2000*; *Pilsl et al., 2016*). Rrn3 association stabilizes Pol I in its monomeric and initiation-competent form (*Blattner et al., 2011*; *Engel et al., 2016*; *Pilsl et al., 2016*; *Torreira et al., 2017*), with which Core Factor further engages to facilitate Pre-Initiation Complex assembly and transcription initiation (*Aprikian et al., 2001*; *Knutson and Hahn, 2013*; *Milkereit and Tschochner, 1998*; *Peyroche et al., 2000*; *Schneider, 2012*). In addition to recruiting

Pol I/Rrn3 to the ribosomal DNA (rDNA) promoter, Core Factor has also been implicated in transcription bubble opening (*Kahl et al., 2000*). Core Factor recruitment to the rDNA promoter in vivo requires the association of UAF with the upstream activating sequence (UAS) and TBP (*Bordi et al., 2001*; *Oakes et al., 1999*; *Steffan et al., 1996*; *Vannini, 2013*).

Transcription initiation by the three eukaryotic RNA RNA polymerases requires transcription factor (TF) IIB-like factors (*Vannini, 2013*; *Vannini and Cramer, 2012*). TFIIB binds the Pol II dock and wall domains using its N-terminal zinc ribbon (ZR) (*Bushnell et al., 2004*; *Chen and Hahn, 2004*) and C-terminal cyclin fold domains (*Chen and Hahn, 2004*; *Kostrewa et al., 2009*; *Sainsbury et al., 2013*), respectively. The Rrn7 subunit of Core Factor is predicted to share sequence homology with TFIIB (*Blattner et al., 2011*; *Knutson and Hahn, 2011*; *Naidu et al., 2011*), containing similar ZR and cyclin fold domains in addition to a helical C-terminal domain (CTD). Models for the Pol I Pre-Initiation Complex were proposed based on the similarity between TFIIB and Rrn7 (*Blattner et al., 2011*; *Knutson et al., 2014*), which have recently been challenged by the crystal structure of Core Factor (*Engel et al., 2017*). In addition to Rrn7, Rrn6 and Rrn11 are essential subunits of Core Factor (*Lalo et al., 1996*; *Lin et al., 1996*). The human ortholog of Core Factor is Selectivity Factor 1 (SL1), which comprises three evolutionarily conserved core subunits and two additional metazoan-specific subunits, TAF1D and TAF12 (*Denissov et al., 2007*; *Gorski et al., 2007*), suggesting a conserved architecture between these Pol I general transcription factors (*Knutson and Hahn, 2013*; *et al., 2006*; *Schneider, 2012*).

Structural approaches have elucidated the dynamic nature of Pol I, possibly reflecting potential conformational states that it can adopt during different stages of transcription. First, in the atomic structure of Pol I determined by X-ray crystallography, a dimeric configuration and an expanded DNA-binding cleft were observed (*Engel et al., 2013*; *Fernández-Tornero et al., 2013*). The DNA-binding cleft is occupied by an element named the expander, mimicking a DNA molecule, while another element called the connector contributes to the dimerization interface by engaging the clamp domain of the neighboring Pol I (*Engel et al., 2013*; *Fernández-Tornero et al., 2013*). In addition, the bridge helix at the active site partially unfolds. When interacting with Rrn3, both the expander and the connector are displaced, resulting in a monomeric form of Pol I, with a more contracted cleft and a partially rewound bridge helix (*Engel et al., 2016*; *Pilsl et al., 2016*). A further contraction of the cleft and a completely folded bridge helix were observed in the elongation form of Pol I revealed by cryo-EM (*Neyer et al., 2016*; *Tafur et al., 2016*). Although these studies provided intriguing hints at the mechanisms of Pol I transcription initiation and elongation, the lack of a Pre-Initiation Complex in these studies precluded a full understanding of its engagement with the promoter and its transition to an active transcribing state.

To gain insight into Core Factor's role during Pol I transcription initiation, we obtained a Pol I Initial Transcribing Complex (ITC) and determined its structure to near-atomic resolution using cryo-EM. In particular, we describe three distinct functional states of the Pol I initiation complexes visualized at 3.8–4.3 Å resolution. Our structures reveal unexpected features of Core Factor's binding to Pol I and promoter DNA compared to the Pol II Pre-Initiation Complex, and provide novel insight into the mechanism of Pol I promoter opening utilizing the intrinsic mobility of Core Factor in the absence of ATP hydrolysis

## Results

### Assembly and cryo-EM reconstruction of the Pol I initiation complex on promoter DNA

To gain insight into the regulation of Pol I transcription initiation, we assembled the Pol I basal transcription complex on an rDNA promoter using purified factors from *Saccharomyces cerevisiae* (Materials and methods; *Figure 1—figure supplement 1*). In order to stabilize the complex, we used a nucleic acid scaffold containing a 17-nucleotide (nt) mismatched transcription bubble in the presence of a 6-nt RNA molecule, mimicking an initial transcribing state (*Figure 1A*).

Single particle analysis using RELION (*Kimanius et al., 2016*; *Scheres, 2012*) produced a reconstruction with an overall resolution of 3.8 Å (FSC = 0.143 criterion) (*Figure 1B*, *Figure 1—figure supplement 2*, *Video 1*). The reconstruction shows a bipartite configuration, with the large module showing clear features of Pol I and the small lobe corresponding to Core Factor. DNA density was

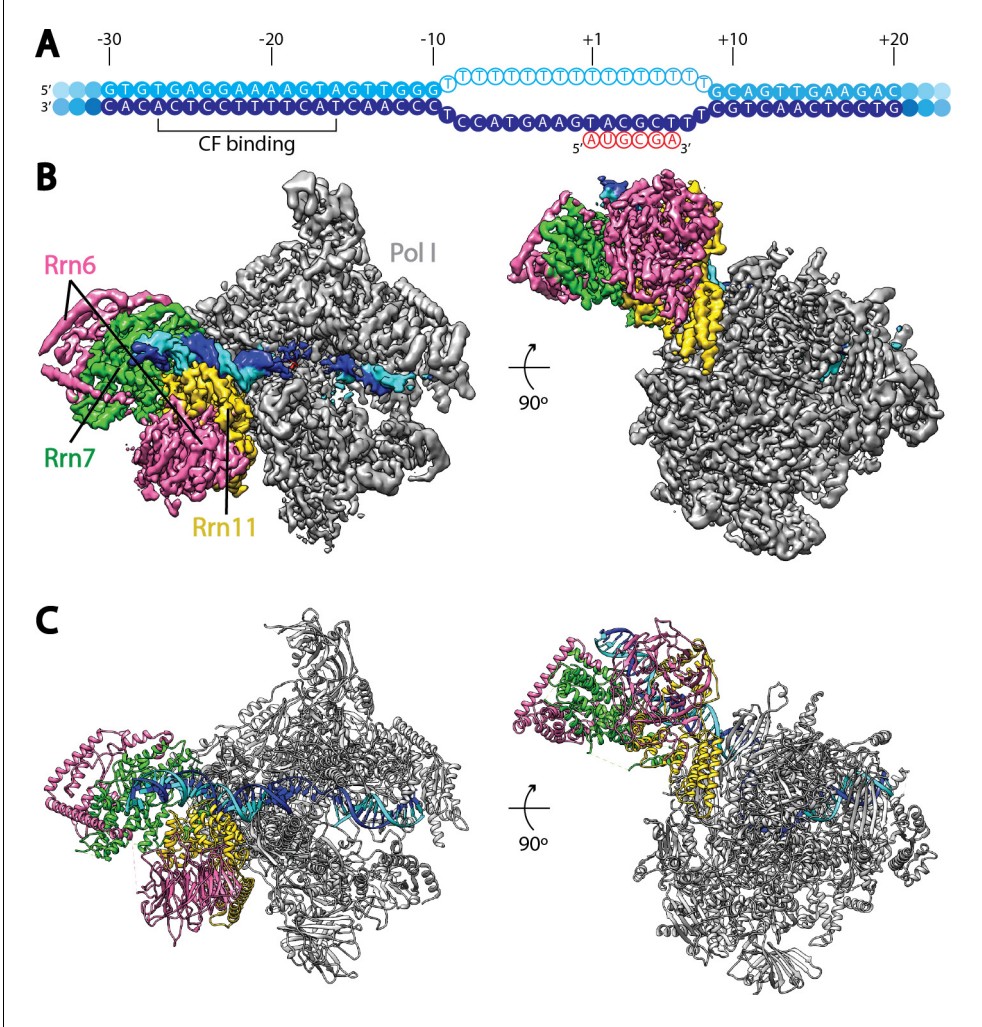

**Figure 1.** Cryo-EM structure of Pol I Initial Transcribing Complex. (**A**) Nucleic acid scaffold used. The non-template and template strands are depicted in cyan and blue, respectively. Filled circles represent rDNA promoter sequence, while open circles show the poly-T mismatch sequences. RNA is shown in red. Core Factor binding region is also labeled. (**B**) Cryo-EM reconstruction of Pol I Initial Transcribing Complex following focused refinements on Core Factor and Pol I separately (Materials and methods). Pol I is colored gray, and nucleic acid template is colored as shown in **A**. The Core Factor subunits are depicted in pink (Rrn6), green (Rrn7) and gold (Rrn11). Two views, front (left) and bottom (right), are shown. (**C**) MDFF (molecular dynamics flexible fitting) model of the Pol I Initial Transcribing Complex. Components are colored the same as in **B**.

The following figure supplements are available for figure 1:

**Figure supplement 1.** Yeast Pol I factors and in vitro transcription assay.

**Figure supplement 2.** Cryo-EM of Pol I Initial Transcribing Complex.

**Figure supplement 3.** Comparison of Pol I in our Initial Transcribing Complex reconstruction with previous published models.

**Figure supplement 4.** Rrn3 does not stably associate with the rest of the complex.

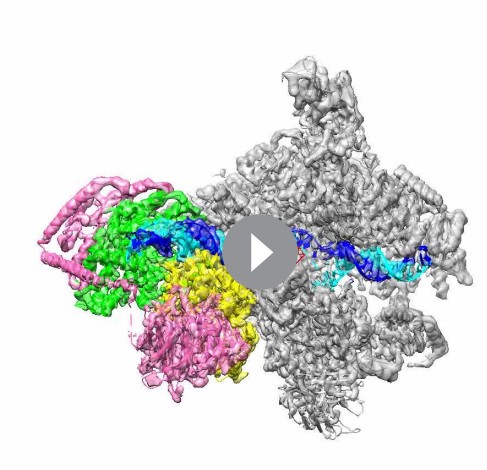

**Video 1.** Cryo-EM reconstruction and MDFF model of Pol I Initial Transcribing Complex. Densities are shown as a semi-transparent surface following a similar color scheme to that in *Figure 1*.

observed both inside and upstream of the Pol I cleft (*Figure 1B*), with the upstream DNA interacting with Core Factor. Local resolution estimation shows that the Pol I core region is very rigid with resolution mostly at 3.5 Å, whereas peripheral regions such as the stalk and Core Factor are more mobile with a lower resolution (4.5 to 5.5 Å) (*Figure 1—figure supplement 2C*). This indicates a flexible nature of the Core Factor in the Initial Transcribing Complex. Indeed, maximum-likelihood based 3D classification revealed that Core Factor adopts different orientations relative to Pol I, likely suggesting a continuous motion for Core Factor. Unique features were also observed accompanying the movement of Core Factor, which will be discussed in a subsequent section.

To obtain a higher resolution density map, especially for the region corresponding to the Core Factor-DNA interaction, we applied separate soft masks around the Core Factor and Pol I densities and performed focused refinements for both individually (*Figure 1—figure supplement 2C*). This resulted in improved density maps for both Pol I and Core Factor, with an overall resolution of 3.7 Å and 4.2 Å, respectively (*Figure 1—figure supplement 2C,D*). This procedure permitted de novo model building for Core Factor (*Figure 1C*). We also generated and refined an atomic model for Pol I based on the crystal structures (*Engel et al., 2013*; *Fernández-Tornero et al., 2013*). Compared to similar structures determined by either crystallography in the apo form (*Engel et al., 2013*; *Fernández-Tornero et al., 2013*) or cryo-EM in an Elongation Complex (EC) (*Neyer et al., 2016*; *Tafur et al., 2016*), Pol I within our Initial Transcribing Complex reconstruction resembles the Elongation Complex more than the apo crystal structure, with a contracted active site cleft (*Figure 1—figure supplement 3*). In addition, Core Factor engages Pol I in the vicinity of the protrusion domain near the upstream entrance of the DNA binding cleft, and intimately interacts with the upstream promoter DNA (*Figure 1B*).

Although we included both Rrn3 and TBP in our assembly reactions (Materials and methods), we did not observe densities in any of the classified reconstructions that could correspond to them (*Figure 1—figure supplement 2*). Therefore, we monitored all fractions of our assembly reaction by gel electrophoresis (*Figure 1—figure supplement 4*). Consistent with the absence of Rrn3 in our reconstructions, we found out that all of the Rrn3 protein was in the unbound and the first wash fractions (lanes 5 and 6 in *Figure 1—figure supplement 4*). Although we cannot rule out the possibility that Rrn3 is present at a substoichiometric level that is below the detection limit of silver staining, or alternatively, that Rrn3 associates with Pol I in the unbound fraction that somehow failed to engage the nucleic acid scaffold, our data suggests that Rrn3 does not stably associate with the rest of Pol I initiation machinery under our experimental conditions. Given the essential roles Rrn3 plays during Pol I transcription (*Keener et al., 1998*; *Moorefield et al., 2000*; *Schneider, 2012*; *Yamamoto et al., 1996*) and its dissociation from Pol I after transcription initiation (*Bier et al., 2004*; *Hirschler-Laszkiewicz et al., 2003*; *Milkereit and Tschochner, 1998*), our data is consistent with the notion that Rrn3 functions at an earlier step during Pol I transcription initiation, where it stabilizes Pol I in an initiation-competent monomeric form and facilitates Pol I recruitment to rDNA promoter. In a separate study, Reeder and colleagues reported that Pol I can be recruited to the promoter in the absence of Rrn3, however this complex is inactive (*Aprikian et al., 2001*), suggesting that Rrn3 may also function post Pol I recruitment. This is also consistent with our structural study, as we included an RNA molecule in our bubble template, which may have resulted in bypassing the requirement for Rrn3 after the engagement of Pol I with the nucleic acid scaffold. As for TBP, we observed a band in the elution fraction that could result from TBP non-specifically binding

to the DNA template (*Figure 1—figure supplement 4*, compare lanes 9 and 12), consistent with the absence of TBP in our structures.

## Molecular structure of Core Factor

The overall structure of DNA-bound Core Factor resembles a right hand holding the DNA molecule between the fingers and the palm, with the thumb pointing toward Pol I (*Figure 2—figure supplement 1A*). The palm is composed of the N-terminal regions of both Rrn11 and Rrn6, the thumb is composed of the C-terminus of Rrn11, and the fingers and knuckles are composed of Rrn7 and the C-terminal half of Rrn6, respectively (*Figure 2—figure supplement 1A*).

Rrn6 plays a scaffolding role in the assembly of Core Factor, spanning the palm and the knuckles (*Figure 2—figure supplement 1A*). As predicted (*Knutson et al., 2014*), Rrn6 is composed of an N-terminal domain (NTD), a WD40 repeat domain, a helical bundle (HB) domain, and a C-terminal unstructured region (*Figure 2A*). The NTD and WD40 domains of Rrn6 reside in the palm where they engage Rrn11 (*Figure 2D*, *Figure 2—figure supplement 1B*), whereas the HB domain forms the knuckles and interacts with Rrn7 (*Figure 2E*). A flexible linker connects the WD40 and HB domains of Rrn6 (*Figure 2A*). No density was observed for the C-terminal region after H9 of the Rrn6-HB domain, in agreement with the lack of structure for this region (*Knutson et al., 2014*).

Rrn11 is predicted to contain a TPR (tetratricopeptide repeat) domain with N- and C-terminal unstructured regions (*Knutson et al., 2014*). In agreement with the prediction, we can assign most of Rrn11 to helical densities, with a total of 13 helices and three long loops (H2-H3, H4-H5, and H8-H9) (*Figures 1B* and *2B*). The C-terminal helices from H5 to H13 resemble a classic TPR domain (*Allan and Ratajczak, 2011*; *D'Andrea and Regan, 2003*) more than H1-H4 (*Figure 2B*). Therefore, we named H5 to H13 the TPR domain of Rrn11, and H1-H4 the NTD. The NTD of Rrn11 resides in the palm (*Figure 2—figure supplement 1A*) and caps the WD40 domain of Rrn6, directly contacting repeats W3 to W5; the TPR domain interacts with repeat W6 and the NTD of Rrn6 (*Figure 2D*, *Figure 2—figure supplement 1B*) forming the thumb (*Figure 2—figure supplement 1A*).

Like its counterpart TFIIB in the Pol II system, Rrn7 contains two cyclin fold domains (*Knutson and Hahn, 2011*; *Naidu et al., 2011*) (*Figure 2C*), which form the fingers (*Figure 2—figure supplement 1A*). The two cyclin fold domains can be aligned with those in TFIIB individually (*Figure 2—figure supplement 2A*), but a different relative orientation between them is adopted compared to their counterparts in TFIIB. One of the cyclin fold domains of TFIIB must be rotated when the other is aligned to its counterpart in Rrn7 (*Figure 2—figure supplement 2B*), to achieve the more compact organization in Rrn7. This compact architecture is presumably induced by Rrn6 intimately embracing Rrn7 (*Figure 2E*). The N-terminal cyclin fold (CyclinN) domain is mainly composed of 5 consecutive α-helices, whereas the C-terminal cyclin fold (CyclinC) domain contains a long insertion between H3 and H4. The insertion region is composed of 5 α-helices referred to as the Insertion Helices (IH) (*Figure 2C*). This finding is consistent with previous sequence analyses, in which the similarity of CyclinC with TFIIB stops at H3 (*Knutson and Hahn, 2011*). The helices H2-H4 of Rrn7 IH contacts Rrn11 TPR H5, H6 and H8, bridging the fingers with the thumb (*Figure 2—figure supplement 1C*).

The interaction between Core Factor and Pol I in the Initial Transcribing Complex is mainly mediated by the Rrn11 TPR domain (the thumb) and the Pol I protrusion (*Figure 2F*). Compared to the large interfaces among Core Factor subunits, the interface between Rrn11 and the Pol I protrusion is rather small, involving Rrn11 helices H8, H10 and H12 (*Figure 2F*). Interestingly, Rrn7 IH H4 is also positioned near Rpb12 subunit of Pol I (*Figure 2F*), possibly contributing to Core Factor/Pol I interaction in the complex. The limited interface between Core Factor and Pol I is consistent with the flexibility of Core Factor observed in our density map (*Figure 1—figure supplement 2C*).

## Pol I and Core Factor are both involved in promoter DNA interactions

The refined structure of the full complex clearly reveals the path of the promoter DNA in the Pol I Initial Transcribing Complex. Densities for both downstream and upstream duplex DNA were clearly resolved (*Figure 1B*). The downstream duplex DNA is inserted into the active site cleft of Pol I, stabilized by interactions with the clamp head, the cleft and the jaw domains of A190, the lobe domain of A135, and Rpb5 (*Figure 3—figure supplement 1*), similar to Pol II initiation complexes (*He et al., 2016*; *Murakami et al., 2015*; *Plaschka et al., 2016*) and Elongation Complexes of all three

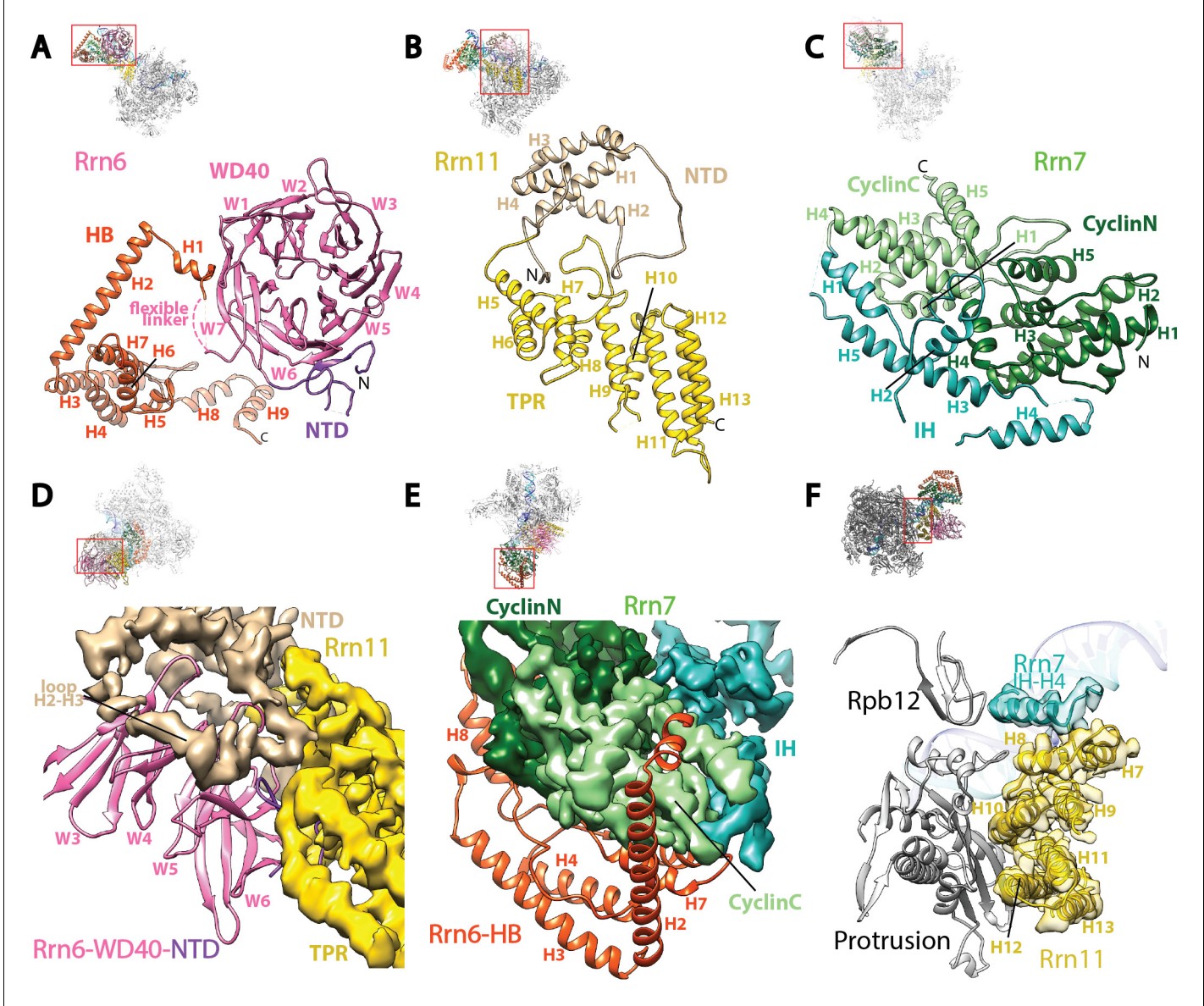

**Figure 2.** Core Factor architecture and Pol I interaction. (A-C) Ribbon diagrams showing the domain architecture of Rrn6 (A), Rrn11 (B), and Rrn7 (C). NTD, N-terminal domain; HB, helical bundle; CyclinC/N, C/N-terminal Cyclin Fold domain; IH, insertion helices; TPR, tetratricopeptide repeats. D and E, Rrn6's scaffolding role in Core Factor assembly by binding Rrn11 (D) and Rrn7 (E) using large interaction surfaces from WD40-NTD and HB, respectively. Color scheme is same as in (A-C). (F) Interface between Core Factor and the Pol I protrusion and subunit Rpb12. Overall views are also shown for each panel, with the same orientation and color scheme. The close-up view is indicated by a red box. Obstructing components are shown in transparency.

The following figure supplements are available for figure 2:

**Figure supplement 1.** Molecular architecture of Core Factor.

**Figure supplement 2.** Structural comparison of the cyclin fold domains between Rrn7 and TFIIB.

eukaryotic RNA polymerases (*Barnes et al., 2015*; *Bernecky et al., 2016*; *Gnatt et al., 2001*; *Hoffmann et al., 2015*; *Neyer et al., 2016*; *Tafur et al., 2016*). The positioning of the downstream duplex DNA upon promoter opening and during active elongation suggests conserved mechanisms

of DNA translocation among all three eukaryotic RNA polymerases, consistent with the fact that all RNA polymerases share a conserved catalytic core complex (*Vannini and Cramer, 2012*).

The resolution of our reconstruction of Core Factor at 4.2 Å hinders us from confidently resolving the register of the upstream promoter sequence. To overcome this, we assembled the Pol I/Core Factor complex using a truncated nucleic acid scaffold (*Figure 3—figure supplement 2A*) and obtained a cryo-EM reconstruction at an overall resolution of 6.9 Å (*Figure 3—figure supplement 2B*). In this scaffold, the non-template strand was truncated to position −27 (transcription start site as +1) from the upstream, and the poly-T mismatch sequences were also removed, resulting in an 18 bp duplex sequence (from −27 to −10) in the upstream region (*Figure 3—figure supplement 2A*). In the reconstruction, we observed a clear shortening of the duplex DNA density from the upstream compared to our high-resolution reconstruction assembled on the full scaffold shown in *Figure 1A* (*Figure 3—figure supplement 2C*). This shortened density fits very well with a model of duplex DNA comprising only 18 base pairs (*Figure 3—figure supplement 2D*). By aligning the two reconstructions together, we were able to confidently assign the sequences in the upstream DNA.

Examination of the upstream promoter interactions in the Pol I Initial Transcribing Complex reveals unexpected features that are distinct from Pol II initiation complexes (*Figure 3*). To prevent the upstream DNA from sliding during promoter melting, a 90° bend is induced in the Pol II Pre-Initiation Complex by the binding of TPB and further stabilized through the TFIIB cyclin folds–BRE (TFIIB recognition element) interaction flanking the TATA box (*Kim et al., 1993a*, *1993b*; *Lee and Hahn, 1995*; *Nikolov et al., 1995*; *Tsai and Sigler, 2000*). In contrast, two consecutive kinks of ~35° and ~45° near position −16 that are approximately 5 bp apart are generated by multiple protein-DNA contacts (*Figure 3A*). This is unlikely due to the absence of TBP in our structure, because TBP does not contact rDNA promoter using its TATA-binding saddle (*Bric et al., 2004*) and there is no consensus TATA-box sequence within the Core Factor binding region. Furthermore, TBP is not required *in vitro* but rather stimulates Pol I transcription (*Aprikian et al., 2000*; *Bedwell et al., 2012*; *Siddiqi et al., 2001*). The Core Factor-DNA interaction reported herein is more similar to the *Acanthameoba castellanii* TIF-IB (transcription initiation factor IB, homologous to yeast Core Factor)-DNA complex, which revealed a ~ 45° bending in the DNA at position −23 (*Gong et al., 1995*).

The interactions between Core Factor and promoter DNA are mainly mediated by subunits Rrn7 and Rrn11 (*Figure 3A*). Rrn7 mainly uses its N-terminal cyclin fold domain to interact with promoter DNA from positions −27 to −20 (*Figure 3B*). This interaction involves helices H3 and H5, as well as the loops H2-H3 and H4-H5, contacting mainly the backbone of the promoter DNA. In addition to these elements, we also observed the loop H2-H3 in the C-terminal cyclin fold domain reaching into the major groove of the DNA near position −27, the furthest interface between Core Factor and DNA (*Figure 3B*). This is likely to be an important interface, because we failed to assemble the Pol I/Core Factor complex when we used a scaffold truncated to position −26 (that is, when the residue in the non-template strand at position −27 was deleted). Rrn11 contacts promoter DNA from positions −24 to −16 (*Figure 3C*). Like the Rrn7-DNA interaction, Rrn11 mainly contacts DNA through backbone interactions, including its NTD helices H1, H2 and H4, as well as H5 in the TPR domain (*Figure 3C*). The extended loop between helices H8 and H9 of the Rrn11 TPR domain is sandwiched between H2 and promoter DNA, contacting the DNA backbone near base-pair −16 (*Figure 3C*). These Core Factor-promoter interfaces agree well with previous photo-crosslinking (*Bric et al., 2004*) and Methidiumpropyl-EDTA·Fe(II) footprinting (*Bateman and Paule, 1988*) experiments.

Pol I also contacts the upstream DNA near position −10 (*Figure 3D*). At this location, the duplex DNA is sandwiched between the wall and the protrusion (*Figure 3D*). This is distinct from Pol II Pre-Initiation Complex, in which the upstream promoter DNA is positioned above the Pol II cleft (*He et al., 2016*; *Murakami et al., 2015*; *Plaschka et al., 2016*). Interestingly, when Pol II enters the elongation state, the upstream duplex DNA shifts ~20 Å (*Barnes et al., 2015*; *Bernecky et al., 2016*) to the same corresponding location as the upstream DNA in the Pol I Initial Transcribing Complex. This, together with the fact that upstream DNA is also stabilized by the A135 protrusion domain in Pol I Elongation Complex (*Tafur et al., 2016*), indicates that Pol I initiation complex is pre-conditioned in an elongation-competent state by Core Factor even at the initiation step. Taken together, the Pol I Initial Transcribing Complex structure reveals a different conformation of the promoter DNA, suggesting distinct initiation mechanisms between Pol I and II.

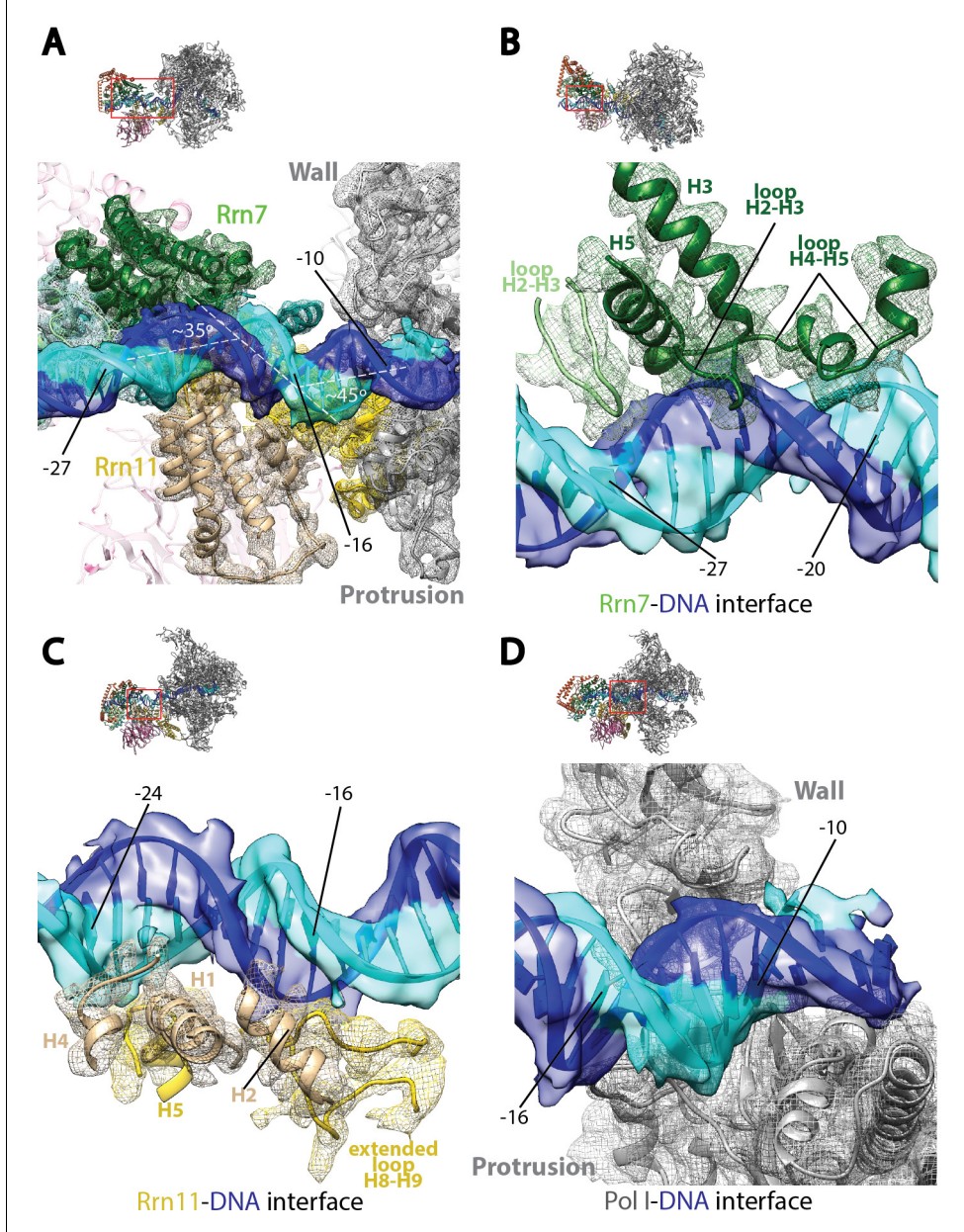

**Figure 3.** Core Factor engagement with promoter DNA from positions −27 to −16. (**A**) Rrn7 and Rrn11, as well as the Pol I wall and protrusion domains interact with rDNA promoter, resulting in two consecutive kinks of ~35 and~45 degrees in promoter DNA approximately 5 bp apart. Rrn7, Key DNA binding domains and promoter DNA are shown as ribbons fitted in their corresponding density (mesh for proteins and transparent surface for DNA). Rrn6 and the remaining of Pol I are shown as transparency in the background. (**B**) Rrn7's interaction with promoter DNA from positions −20 to −27. (**C**) Rrn11's contact rDNA promoter from positions −16 to −24. (**D**) Sandwiching of promoter DNA near position −10 by the wall and protrusion. Overall views are also shown for each panel, with the same orientation and color scheme. The close-up view is indicated by a red box. Obstructing components are shown in transparency.

The following figure supplements are available for figure 3:

**Figure supplement 1.** Interfaces for the downstream duplex DNA in the Pol I Initial Transcribing Complex.

**Figure supplement 2.** Cryo-EM reconstruction of Pol I Initial Transcribing Complex using a truncated scaffold.

## Three functional states reveal conformational changes in Pol I Initial Transcribing Complex during transcription initiation

To gain insight into the nature of Core Factor flexibility, we performed 3D refinement on three different classes that were produced by the second 3D classification step of our data processing procedure (Materials and methods; *Figure 1—figure supplement 2C*). We thereby subsequently

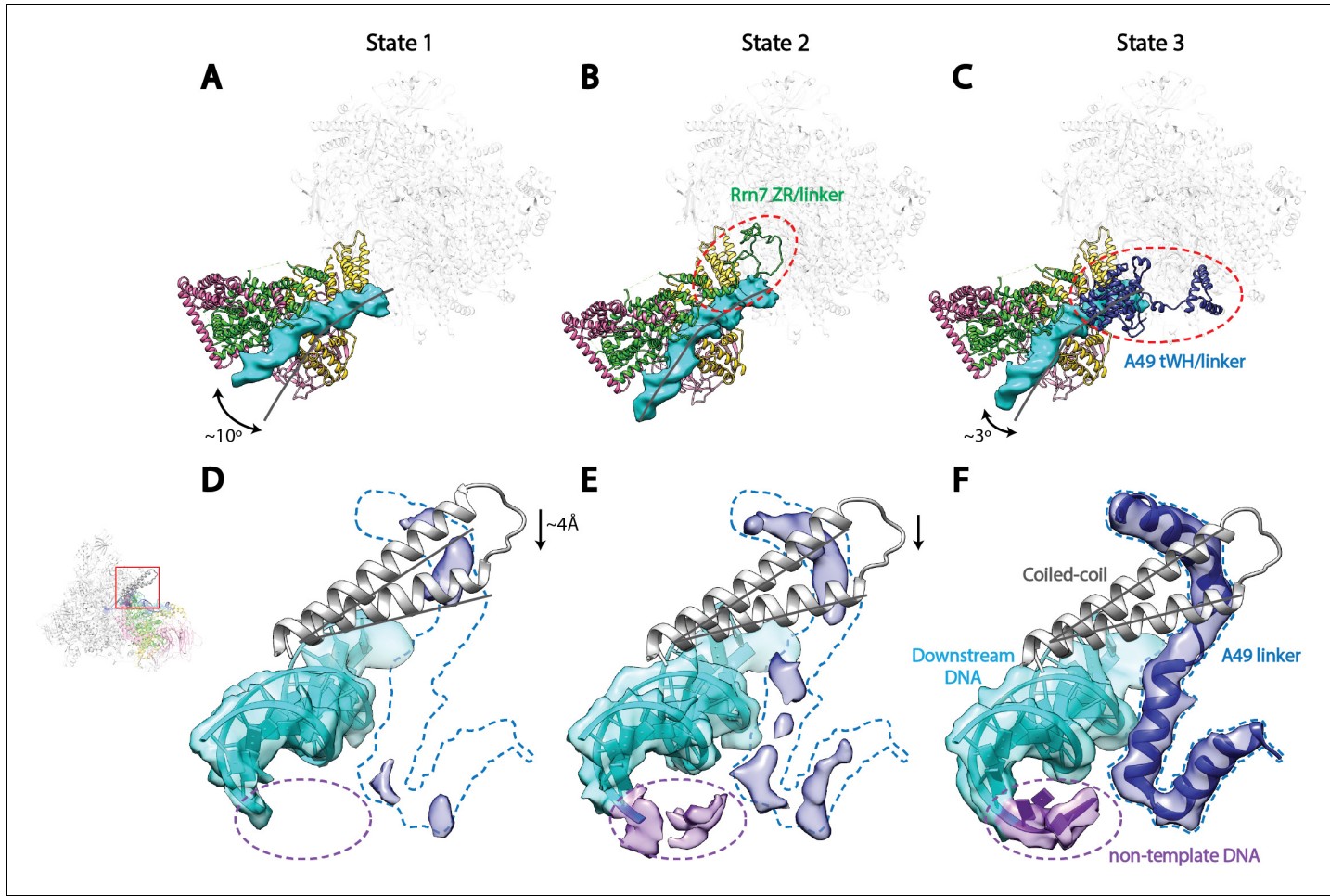

**Figure 4.** Correlation of structural states with key functional elements in Pol I Initial Transcribing Complex. (**A-C**) Top-down views showing the location of the Core Factor/DNA complex relative to Pol I in State 1 (**A**), State 2 (**B**), and State 3 (**C**). The gray lines represent the path of upstream DNA in State 2, in which Core Factor moves closest to Pol I. Dotted red circles denote the locations of the Rrn7 N-terminal ZR and linker regions (**B**) and the A49 tandem winged helix (tWH) and linker domains (**C**). Rotations of ~10° and ~3° for the Core Factor/DNA complex by comparing State 1 and 3 with State 2 is also labeled in **A** and **C**. (**D-F**) Structural mobility of the active site cleft in State 1 (**D**), State 2 (**E**), and State 3 (**F**). The coiled-coil is shown as gray ribbon, with gray lines marking the position of the coiled-coil in State 3, where the clamp domain adopts the most contracted conformation. Arrows in **D** and **E** indicate the movement of the coiled-coil compared to State 3 (**F**). The A49 linker and downstream duplex DNA are shown as transparent density fitted with ribbon models in blue and cyan, respectively. Dotted outline in medium blue and purple represent the locations of the A49 linker and the non-template DNA, respectively. Zoom out view for **D-F** is shown to the left of panel **D**.

The following figure supplements are available for figure 4:

**Figure supplement 1.** Structural features of the three functional states of Pol I Initial Transcribing Complex.

**Figure supplement 2.** Comparison of the density of the template strand within the active site cleft among the three functional states of the Pol I Initial Transcribing Complex.

**Figure supplement 3.** Core Factor mutants are defective for open complex formation.

obtained reconstructions of the Pol I Initial Transcribing Complex in three distinct functional states (*Figure 4*, *Figure 4—figure supplement 1*), with overall resolutions of 4.2 Å, 4.3 Å and 3.9 Å for states we respectively designate State 1, 2, and 3. Further examination of the three states reveals unexpected conformational changes correlating with the position of Core Factor (*Figure 4*).

Core Factor and upstream DNA adopt a range of positions on Pol I, with a rotational movement of up to 10° (*Figure 4A–C*). State 1 shows a striking feature that the upstream DNA and Core Factor are rotated more toward the back of Pol I, projecting the DNA away from the active site (*Figure 4A*). Compared to State1, the Core Factor/DNA complex is rotated ~10° toward the front of Pol I in State 2 (*Figure 4B*), allowing the upstream promoter to insert into the active site cleft, whereas the Core Factor/DNA complex resides at an intermediate location in State 3 (*Figure 4C*). Because the contact point between Core Factor and Pol I is relatively small (*Figure 2F*), this degree of movement can be readily achieved. Indeed, examination of the Core Factor/Pol I interface shows that Core Factor and the upstream DNA pivot around the position where Rrn11 H8 interacts with the Pol I protrusion domain (*Figure 4—figure supplement 1A*), suggesting Rrn11 H8 may be functionally important. Indeed, previous work has shown that removal of Rrn11 H7-8 results in a lethal growth phenotype, although Core Factor can still assemble, as would be expected for a Pol I interaction surface (*Knutson et al., 2014*).

Key structural elements in both Core Factor and Pol I are stabilized in distinct functional states. In State 2, the movement of Core Factor/DNA correlates well with the stabilization of Rrn7 zinc ribbon/linker on Pol I (*Figure 4B*, *Figure 4—figure supplement 1B*). This likely positions the Rrn7 linker region close to the single-stranded template DNA in the cleft, suggesting that the Rrn7 zinc ribbon/linker region might play a role similar to that of TFIIB during Pol II transcription initiation. Indeed, yeast extracts prepared from the strain lacking the zinc ribbon domain of Rrn7 (ΔZR) failed to support the open complex formation (*Figure 4—figure supplement 3A,B*). In addition, the recombinant ΔZR Core Factor also failed to recover the defect in open complex formation in the ΔZR extract, whereas the wild type (WT) complex did (*Figure 4—figure supplement 3B,C*). This data, together with the finding that Core Factor complex bearing the ΔZR Rrn7 can still be recruited to rDNA promoter (*Knutson et al., 2014*), indicates that the Rrn7 N-terminal region promotes transcription bubble opening, consistent with the State 2 structure. Intriguingly, in State 3, we observed strong density near the Pol I wall domain and the upstream promoter DNA corresponding to the A49 tandem winged helix (tWH) domain (*Figure 4—figure supplement 1C*). In addition, the A49 linker connecting tWH to the A49/A34.5 dimerization domain was also resolved, which spans the active site cleft and contacts the coiled-coil (*Figure 4F*). Comparison of State 2 with State 3 reveals another striking feature in that there are clear clashes between the Rrn7 linker region and the A49 tandem winged helix domain (*Figure 4—figure supplement 1D*). This suggests that in addition to its function in transcription elongation, the A49 tandem winged helix domain could also play a role during initiation by displacing the Rrn7 zinc ribbon/linker region, thus opening the RNA exit channel and priming Pol I for promoter escape.

We also discovered a continuous movement of the Pol I clamp, which gradually closes up to 4 Å from State 1 to State 3 (*Figure 4D–F*). Importantly, the closing of the clamp correlates well with the stabilization of the A49 linker region (*Figure 4D–F*) which likely leads to the stabilization of the tandem winged helix domain. This correlation suggests a role for the A49 linker in sensing the width of the active site cleft. In addition, at least three consecutive nucleotides of the non-template DNA at the downstream fork of the transcription bubble is also gradually stabilized (*Figure 4D–F*). Interestingly, we also observed subtle but distinct differences for the density of the template strand DNA (*Figure 4—figure supplement 2*) correlated with this movement.

In summary, we observed that the position of Core Factor/DNA lobe correlates with the stabilization of key elements such as the Rrn7 zinc ribbon/linker, A49 tandem winged helix/linker, and Pol I clamp during transcription initiation, suggesting Pol I-specific initiation mechanisms which have not been observed from the Pol II system (see Discussion).

## Discussion

Engagement of upstream promoter DNA with general transcription factors is an essential step during transcription initiation. For Pol II, large conformational changes and species-specific features have been observed from the closed complex formation to active elongation (*Han and He, 2016*;

*He et al., 2016*; *Plaschka et al., 2016*). We have reported here cryo-EM structures of Pol I Initial Transcribing Complex captured in three different functional states (*Figure 4*). During the preparation of this manuscript, a study on the same topic was published using a combination of X-ray crystallography and cryo-EM showing the unique feature of Pol I/Core Factor interaction and their functional relevance (*Engel et al., 2017*). In this study, we have captured three functional states of the Pol I Initial Transcribing Complex for the first time, revealing a more dynamic picture of Pol I transcription initiation and its transition to an actively transcribing state.

## Comparison between Pol I and Pol II Pre-Initiation Complexes

Comparison with the better characterized Pol II complexes suggests both shared as well as distinct mechanisms used by Pol I for transcription initiation (*Figure 5*). First, TFIIB-like factors are required for the transcription initiation of all three eukaryotic RNA polymerases (*Vannini and Cramer, 2012*). Rrn7 is the Pol I-specific TFIIB-like factor (*Knutson and Hahn, 2011*; *Naidu et al., 2011*). In our reconstruction of State 2, we observed the Rrn7 zinc ribbon domain residing in the RNA exit channel, similar to that of TFIIB (*Figure 5—figure supplement 1A*), consistent with its functional conservation based on similarity at the level of domain architecture. However, the cyclin fold domains of Rrn7 are located at a more distal position relative to Pol I, different from that of TFIIB (*Figure 5—figure supplement 1A*). Consistent with this, the cyclin fold domains of Rrn7 are mainly involved in promoter DNA interaction (*Figure 3*), lacking an essential interface with the polymerase compared to TFIIB. This function is somewhat replaced by the Rrn11 subunit of Core Factor. In addition, the N-terminal cyclin fold domain in TFIIB specifically recognizes the downstream BRE (TFIIB recognition element), forcing the DNA to take a completely different path (*Figure 5—figure supplement 1B*). Therefore, the TFIIB-like factors play distinct roles in Pol I and Pol II systems.

Second, the upstream promoter DNA in Pol I Initial Transcribing Complex and Elongation Complex occupy a very similar location on Pol I by contacting both the wall and the protrusion (*Figure 5A*), implying that Pol I is pre-conditioned in an elongation-competent form at the initiation stage. Moreover, our Initial Transcribing Complex structures suggest a ratcheting movement of the upstream DNA induced by a flexible Core Factor-Pol I association. The positioning of the upstream promoter DNA in the Pol I Initial Transcribing Complex is fundamentally different from that in the Pol II Pre-Initiation Complex (*Figure 5B,C*). Although the upstream DNA in Pol II Elongation Complex occupies a similar location, it is positioned over the cleft in the Pol II Closed Complex (CC) and

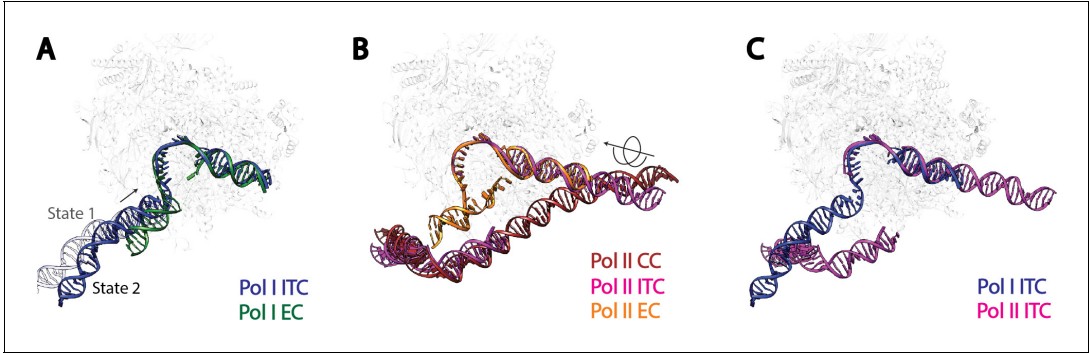

**Figure 5.** Comparison of DNA trajectories among Pol I and Pol II Pre-Initiation Complexes. Top-down views showing the DNA paths of Pol I Initial Transcribing Complex (ITC) in this study and Elongation Complex (EC) (**A**), Pol II Closed Complex (CC), Initial Transcribing Complex and Elongation Complex from previous studies (**B**), Pol I Initial Transcribing Complex and Pol II Initial Transcribing Complex in comparison (**C**). Aligned polymerases are shown as transparent background. DNA is shown in blue for Pol I Initial Transcribing Complex; in green for Pol I Elongation Complex; in brown for Pol II Closed Complex; in magenta for Pol II Initial Transcribing Complex; in orange for Pol II Elongation Complex. Structural models used: Pol I Elongation Complex, 5M5X (*Tafur et al., 2016*); Pol II Closed Complex, 5FMF (*Murakami et al., 2015*); Pol II Initial Transcribing Complex, 4V1N (*Plaschka et al., 2015*) and 5IYD (*He et al., 2016*); Pol II Elongation Complex, 5C4X (*Barnes et al., 2015*).
The following figure supplement is available for figure 5:

**Figure supplement 1.** Comparison of Pre-Initiation Complexes of Pol I and II near TFIIB-like proteins.

the Initial Transcribing Complex (*Figure 5B*). Transition from initiation to elongation for Pol II requires the ATP-dependent translocase activity of TFIIH to insert DNA from downstream into the active site cleft (*Fishburn et al., 2015*; *Grünberg et al., 2012*; *He et al., 2013*; *Kim et al., 2000*). However, TFIIH is not required for Pol I transcription initiation (*Assfalg et al., 2012*), and no ATP hydrolysis activity has been associated with Pol I general transcription complexes (*Lofquist et al., 1993*). Instead, we observed three functional states of the Pol I Initial Transcribing Complex (*Figure 4*), which allowed us to propose a model shedding light on the mechanisms of Pol I transcription initiation in the absence of ATP-hydrolysis (*Figure 6*; *Video 2*).

## Temporal steps during Pol I transcription initiation

Transcription initiation by Pol I is a dynamic process, and the three functional states of Pol I Initial Transcribing Complex likely represent snapshots of ordered events during transcription initiation: State 1 precedes State 2, which is then followed by State 3. Our reasoning is as follows. First, we observed a gradual closing of the cleft from State 1 to State 3. This is in accordance with the findings that the Pol I cleft is in an open configuration when not bound to DNA as is observed in the apo crystal structures (*Engel et al., 2013*; *Fernández-Tornero et al., 2013*), which contracts upon DNA engagement in Elongation Complex (*Neyer et al., 2016*; *Tafur et al., 2016*). Thus, State 1 with a slightly open cleft should correspond to the step right after the loading of the promoter DNA, whereas State 3 adopts a more closed cleft resembling the elongation mode. Second, in line with the movement of the clamp, the A49 tandem winged helix domain is only stabilized in State 3, along with the gatekeeper helices in the linker region spanning over the cleft (*Figure 4C,F*). Given the roles of A49 tandem winged helix domain during transcription elongation (*Beckouet et al., 2008*; *Geiger et al., 2010*; *Pilsl et al., 2016*), State 3 may reflect the initial transcribing state prior to promoter escape. Third, State 2 is an intermediate between State 1 and 3 in terms of clamp closing and tandem winged helix domain stabilization (*Figure 4D–F*). Additionally, we observed stabilized zinc ribbon and linker regions of Rrn7, in a conformation similar to that of TFIIB (*Figure 5—figure supplement 1A*). Moreover, the Rrn7 zinc ribbon domain functions in transcription bubble opening (*Figure 4—figure supplement 3*). Thus, State 2 may represent the state of bubble opening, following promoter DNA insertion while preceding initial RNA synthesis. In summary, the three distinct functional states of the Pol I Initial Transcribing Complex revealed in a single sample provide a temporal view of the dynamic processes of Pol I transcription initiation.

## Mechanisms of Pol I transcription initiation

To provide insight into the mechanisms of Pol I transcription initiation, we first generated a model for the Pol I Closed Complex. As stated above, State 1 likely represents a step immediately after promoter DNA insertion. Therefore, we modeled the Closed Complex based on our reconstruction in State 1 by extending the upstream DNA into the active site cleft of Pol I using B-form DNA. Interestingly, when the upstream DNA is extended into the cleft, no obvious clash between Pol I and DNA is observed (*Figure 6—figure supplement 1*). Therefore, we speculate that this DNA path and position of Core Factor could represent that of a closed complex (*Figure 6A*; *Video 2*). In agreement with this, the Pol I cleft is in a slightly open configuration (*Figure 4C*), presumably accommodating the insertion of the promoter DNA. Thus, the promoter DNA is positioned closer to the Pol I active site than that in the Pol II Pre-Initiation Complex (*He et al., 2016*; *Murakami et al., 2015*; *Plaschka et al., 2016*, *2015*). Meanwhile, the Core Factor is brought closest to Pol I and thereby the downstream DNA is under the most tension by contacting Rpb5. Thus, the intrinsic mobility of Core Factor may constantly insert the duplex DNA toward the active site by applying a force against Rpb5 (*Figure 6A*). When Core Factor reaches a critical position, the Rrn7 ZR/linker regions are stabilized on Pol I. This likely results in a spontaneous melting of DNA that is stabilized by the closing Pol I active site cleft and Rrn7 N-terminal region, followed by the stabilization of the downstream DNA by the clamp head and jaw domains of A190, the lobe domain of A135, and Rpb5, forming a stable open complex (*Figure 6B*; *Video 2*). This model is consistent with the finding that rDNA promoter opening only requires TIF-IB/CF and Pol I in *Acanthamoeba castellanii* (*Kahl et al., 2000*). Next, Pol I initiates RNA synthesis, and the A49 tandem winged helix domain engages near the wall domain on Pol I, presumably stabilizing the upstream DNA. Meanwhile, the transcription bubble within the cleft can also be restricted by the A49 gatekeeper helices in the linker region (*Figure 6C*; *Video 2*).

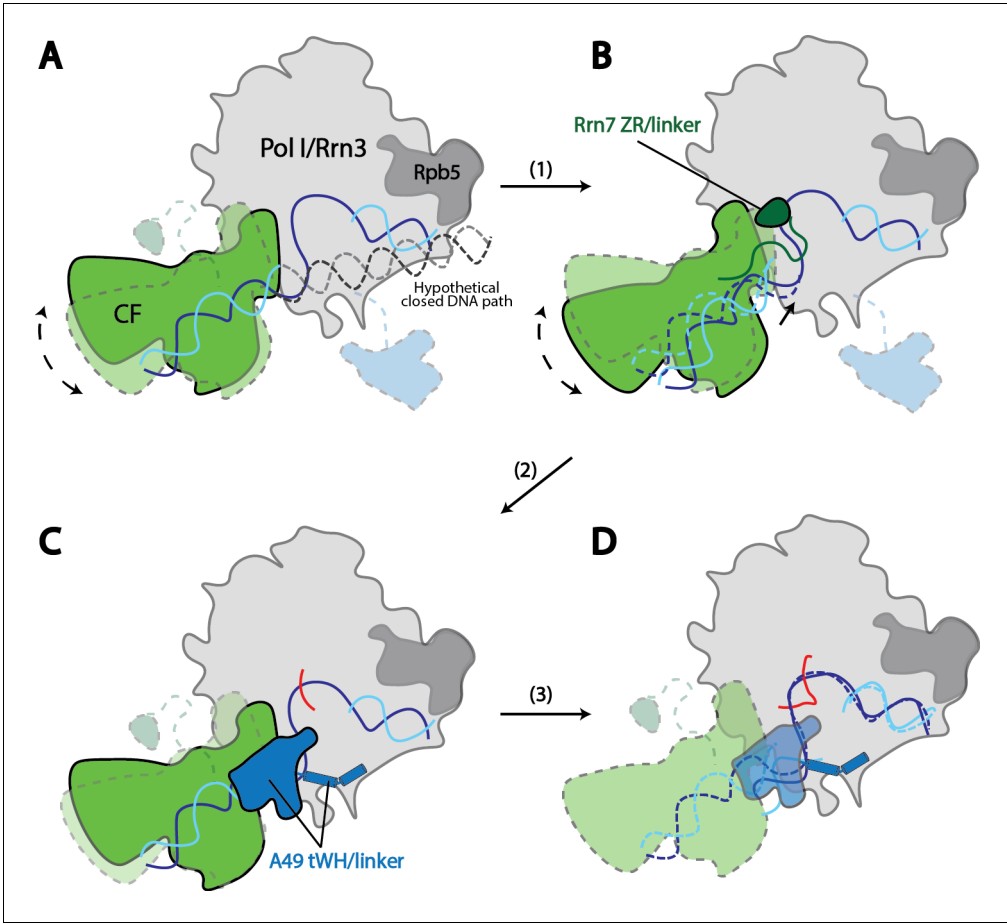

**Figure 6.** Model for Pol I transcription initiation.  Promoter bound Core Factor recruits Pol I/Rrn3, and loads DNA into the cleft of Pol I (**A**). At this stage, both Rrn7 N-terminal domain and A49 tandem winged helix domain are flexible. Pol I cleft is open to make room for loading of DNA. The intrinsic mobility of Core Factor upon Pol I engagement then ratchets upstream DNA against Rpb5. Promoter DNA melting occurs when Core Factor reaches a critical position where Rrn7 zinc ribbon/linker regions are stabilized on Pol I (1). This is likely a very transient state, allowing Pol I to recognize the initiation site in the template strand DNA and start synthesizing the RNA (**B**). Besides the well-accepted role for sensing the growing RNA length for its counterpart TFIIB in the Pol II PIC, Rrn7 likely plays an additional role in facilitating promoter opening by reaching into the RNA exit channel of Pol I and favoring the bending of DNA. During further translocation of Pol I along the promoter (2), the enzyme is acting more similarly as an elongation mode, with a more closed clamp. The gatekeeper linker helices in A49 between the dimerization and tandem winged helix domains presumably work as a ruler of the active site cleft, stabilizing upon clamp closing down while preventing escape of the downstream DNA (**C**). As RNA grows longer, the A49 tandem winged helix domain can also help displacing the Rrn7 zinc ribbon to clear the RNA exit channel. Subsequently, Pol I escapes the promoter and enters a processive elongation state (3). Pol I in its active, Rrn3 bound form is shown in gray, and Core Factor in green. Transparency and dotted outline indicate flexibility. Solid blue and cyan lines depict the DNA paths that has been experimentally observed [panels **A–C** revealed in this study; panel **D** shown in previous studies (*Neyer et al., 2016*; *Tafur et al., 2016*), while dotted blue and cyan lines in **B** and **C** represent the DNA paths in the preceding stages. In panel **A**, a hypothetical closed DNA path is depicted as dotted black/gray lines by naturally extending a B-form DNA from upstream Core Factor associated DNA. RNA molecule with growing length during initial stages of transcription is represented in red.

The following figure supplement is available for figure 6:

**Figure supplement 1.** Model for the proposed Closed Complex based on the reconstruction in State 1.

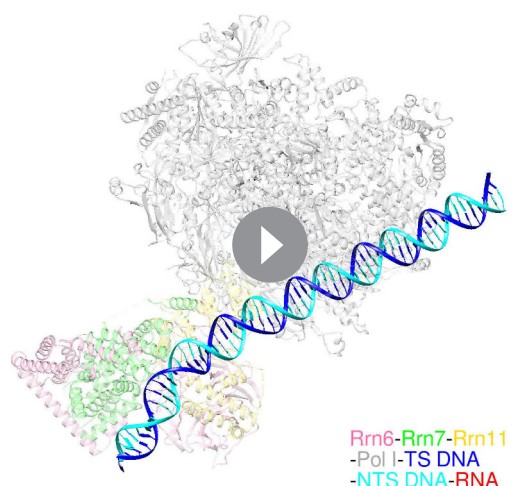

Rrn6-Rrn7-Rrn11
-Pol I-TS DNA
-NTS DNA-RNA

**Video 2.** Molecular model for Pol I transcription initiation. A morph among the modeled Closed Complex (CC), modeled Open Complex (OC) and Initial Transcribing Complex (ITC) states highlights the potential mechanism for promoter opening during Pol I transcription initiation. Key conformational changes observed in the three distinct functional states of the Pol I Initial Transcribing Complex are also depicted. The color scheme is same as in *Figure 1*.

Intriguingly, the A49 tandem winged helix domain can also displace the Rrn7 zinc ribbon/linker to clear the RNA exit channel for subsequent RNA extension (*Figures 4B* and *6C*; *Video 2*). When the nascent RNA extends to a certain length, Pol I escapes the promoter and enters elongation (*Figure 6D*). Our model provides new insight into the mechanisms of promoter opening by Pol I that does not require the ATP-dependent translocase activity of TFIIH, but instead leverages the intrinsic mobility of Core Factor.

Taken together, we have determined the structure of Pol I transcription initiation complex at near-atomic resolution using single particle cryo-EM. The structure reveals the architecture of Core Factor binding to promoter DNA and that Pol I and promoter DNA are pre-conditioned in an elongation-competent form. We have also obtained three functional states of the Pol I Initial Transcribing Complex, which allows us to propose a molecular mechanism in which Pol I utilizes the intrinsic mobility of the DNA-bound Core Factor in the process of promoter opening. This model explains why TFIIH is not necessary for Pol I promoter opening, and implies that a similar mechanism could also be used by Pol III, the other TFIIH-independent RNA polymerase.

# Materials and methods

## Purification of yeast Pol I

Cells of yeast strain containing a 3× FLAG tag at the C-terminus of A135 were grown at 30°C in YPD (2% w/v glucose, 0.002% w/v adenine) to an $OD_{600}$ of 1.0, and harvested by centrifugation. Cell pellets were washed with Extraction Buffer (100 mM HEPES pH 7.9, 250 mM ammonium sulfate, 1 mM EDTA, and 10% glycerol) supplemented with 0.5 mM dithiothreitol (DTT) and protease inhibitors (1 mM phenylmetholsulfonyl fluoride, 2 mM benzamidine, 3 μM leupeptin, 2 μM pepstatin, 3.3 μM chymostatin), and then stored at −80°C. Approximately 200 grams of cells were thawed and resuspended in 1 ml per gram of IP buffer (Extraction buffer with 0.1% Tween) supplemented with 0.5 mM DTT and protease inhibitors. The cells were lysed using a Bead Beater instrument (BioSpec, Bartlesville, OK) using 425–600 μm glass beads (Sigma, St. Louis, MO). The extract was clarified by centrifugation at 4°C for 30 min at 20,000 × g. The clarified extract was passed through cheese cloth and then added to 2 ml anti-flag agarose beads (Biotool, Houston, TX) and incubated at 4°C for 4 hr. Protein bound beads were washed three times with IP buffer and then poured into an empty column. Bound proteins were eluted with IP buffer containing 300 μg/ml 3× FLAG peptide (Biotool). Peptide eluted Pol I was then further purified over HiTrap Heparin HP column (GE Healthcare, Chicago, IL) using a linear gradient of Buffer A (50 mM HEPES pH 8.0, 200 mM KCl, 5 mM MgCl₂, 0.1 mM EDTA, 5% glycerol, 1 mM TCEP) to Buffer B (Buffer A with 800 mM KCl) over 10 column volumes. Flag-Pol I was eluted between 400–600 mM KCl and peak fractions were desalted in Buffer A, concentrated using Amicon Ultra-100K filter (EMD Millipore, Billerica, MA), and then passed over an S300-HR column (GE Healthcare) in Buffer A. Peak fractions were concentrated again as above and then stored at −80°C.

## Purification of Core Factor

Core Factor was expressed as previously described (*Bedwell et al., 2012*; *Knutson et al., 2014*) with the following modifications. Briefly, Core Factor was expressed from the pET-CF vector containing His$_6$-Rrn7-Rrn11-His$_6$-Rrn6 in BL21-CodonPlus(DE3)-RIL cells. Recombinant Core Factor protein was expressed in 2× Autoinducing Terrific Broth (0.024% w/v tryptone, 0.048% yeast extract w/v, 0.4% v/v glycerol, 17 mM KH$_2$PO$_4$, and 72 mM K$_2$HPO$_4$) supplemented with 20 ml per liter ZY-5052 (25% v/v glycerol, 2.5% w/v glucose, and 10% w/v alpha lactose monohydrate). Inoculated media was grown to an OD$_{600}$ of 0.6 then shifted to 24°C overnight. Cells were harvested by centrifugation, pellets were washed in Tris-buffered saline (50 mM Tris-HCl pH 7.6, 150 mM NaCl), and stored at −80°C. Approximately 100 grams of cells were thawed and resuspended in 5 ml per gram of Extraction Buffer (50 mM HEPES pH 8.0, 500 mM KCl, 20 mM Imidazole, 5 mM MgCl$_2$, 0.1 mM EDTA, 20% glycerol) and supplemented with 1 mM DTT and protease inhibitors (as described above). The cells were lysed using a French Press by passing ~30 ml of resuspended cells through the press twice (~16,000 PSI) with a brief sonication pulse between and after the passes to shear genomic DNA and to reduce viscosity. The extract was clarified by centrifugation at 4°C for 30 min at 20,000 × g. The clarified extract was added to Ni-NTA Sepharose beads (Biotool) and incubated at 4°C for 4 hr in batch. Protein bound beads were washed four times with high salt Wash Buffer (Extraction Buffer but with 1 M KCl) and two times with low salt Wash Buffer (Extraction Buffer but with 200 mM KCl). Bound proteins were eluted with 10 ml of Elution Buffer (50 mM HEPES pH 8.0, 200 mM KCl, 200 mM Imidazole, 5 mM MgCl$_2$, 0.1 mM EDTA, 20% glycerol). Eluted Core Factor was then further purified over HiTrap Heparin HP column (GE Healthcare) using a linear gradient of Buffer A (50 mM HEPES pH 8.0, 200 mM KCl, 5 mM MgCl$_2$, 0.1 mM EDTA, 5% glycerol) to Buffer B (Buffer A with 1 M KCl) over 10 column volumes. Core Factor was eluted between 800–1000 mM KCl. Peak fractions were desalted in Buffer C (50 mM HEPES pH 8.0, 0.1 mM EDTA, 5% glycerol, 0.1% Tween) supplemented with 0.05% tergitol and concentrated using Amicon Ultra-100K filters (EMD Millipore). Concentrated Core Factor was then further purified over a HiTrap Q HP column (GE Healthcare) using a linear gradient of Buffer A to Buffer B (as described above). Protein eluted between 400–600 mM KCl and peak fractions were stored at −80°C.

## Purification of TBP

TBP was expressed from pRSF-His$_6$-TBP in BL21-CodonPlus(DE3)-RIL cells. Recombinant TBP protein was expressed as described above (see Core Factor purification). Approximately 40 grams of cells were thawed and resuspended in 5 ml per gram of Extraction Buffer (as described in Core Factor purification) supplemented with 1 mM DTT, protease inhibitors (described above), and 1 mg/ml of lysozyme. The cells were lysed by sonication. The extract was clarified by centrifugation at 4°C for 30 min at 20,000 × g. The clarified extract was added to Ni-NTA Sepharose beads (Biotool) and purified as described in Core Factor purification. Eluted TBP was concentrated using Amicon Ultra-10K filters (EMD Millipore) and further purified over HiTrap Heparin HP column (GE Healthcare) using a linear gradient of Buffer A to Buffer B over 10 column volumes. TBP eluted between 500–700 mM KCl and peak fractions were pooled and stored at −80°C.

## Purification of Rrn3

Rrn3 was expressed from pCDF-His$_6$-Rrn3 in BL21-CodonPlus(DE3)-RIL cells in Autoinducing Terrific Broth (0.006% w/v tryptone, 0.012% yeast extract, 0.4% v/v glycerol, 17 mM KH$_2$PO$_4$, and 72 mM K$_2$HPO$_4$) supplemented with 20 ml per liter ZY-5052 (25% v/v glycerol, 2.5% w/v glucose, and 10% w/v alpha lactose monohydrate). Inoculated media was grown to an OD$_{600}$ of 0.6 then shifted to 24°C overnight. Cells were harvested by centrifugation, washed with Rrn3 Extraction Buffer (50 mM HEPES pH 8.0, 500 mM KCl, 20 mM Imidazole, 5 mM MgCl$_2$, 0.1 mM EDTA, 20% glycerol, 0.1% Tween) supplemented with 1 mM DTT and protease inhibitors (described above) and pellets were stored at −80°C. Thawed cells were resuspended in 5 ml per gram of supplemented Rrn3 Extraction Buffer containing 0.1 mg/ml lysozyme and incubated on ice for 30 min followed by sonication. Lysed cells were clarified by centrifugation and then incubated with Ni-NTA Sepharose beads (Biotool) for 4 hr at 4°C in batch. Protein bound beads were washed four times with Rrn3 Wash Buffer (50 mM HEPES pH 8.0, 20 mM Imidazole, 5 mM MgCl2, 0.1 mM EDTA, 10% glycerol, 0.1% Tween) containing 1 M KCl and twice with Rrn3 Wash Buffer containing 200 mM KCl. Bead bound proteins were

eluted with Rrn3 Elution Buffer (Rrn3 Wash Buffer containing 300 mM Imidazole). Peak elutions were pooled, buffer exchanged in Rrn3 Wash Buffer, and then repurified over a HisTrap HP column (GE Healthcare) using a linear gradient of Buffer A (50 mM HEPES pH 8.0, 200 mM KCl, 5 mM MgCl$_2$, 0.1 mM EDTA, 5% glycerol, 1 mM TCEP) to Buffer B (Buffer A with 800 mM KCl) over 10 column volumes. Peak fractions were pooled and desalted using an Amicon Ultra-10K filter (EMD Millipore) and then further purified over a HiTrap Butyl HP column (GE Healthcare) using a linear gradient of Buffer A supplemented with 1.5 M ammonium sulfate to Buffer B without ammonium sulfate over 20 column volumes. Peak fractions were pooled, desalted in Buffer A and then further purified on a HiTrap Q HP column using a linear gradient of Buffer A to Buffer B. Peak fractions were desalted and concentrated as above and then stored at −80°C.

## Pol I in vitro transcription

Equal molar ratios (~10 nM) of purified factors and Pol I reporter plasmid (200 ng) were assembled on ice, mixed, and then incubated at 30 degrees for 30 min. Pol I transcripts were processed as previously described (*Knutson et al., 2014*; *Schultz et al., 1991*).

## Open complex assays

Open complex formation was performed as previously described (*Hahn and Roberts, 2000*; *Sasse-Dwight and Gralla, 1991*) with a few minor modifications. Transcription reactions were formed as described for the in vitro transcription assays except 15 μg of indicated extract and 50 ng of rDNA reporter plasmid template was used and DTT and RNase inhibitor were omitted. After 30 min at room temperature, KMnO$_4$ was added to a final concentration of 10 mM and incubated for 2 min and then stopped by 3 μl of 2-mercaptoethanol followed by 180 μl of transcription stop mix. Reactions were extracted with phenol/chloroform and ethanol precipitated. Modified DNAs were resuspended in water and were used as templates for primer extension with Taq master mix (New England Biolabs, Ipswich, MA) using a LacI primer labeled with Cyanine 5.5 on the 5' end. The following thermocycler conditions were used: 95°C for 2 min, then 18 cycles of 95°C for 30 s, 55°C for 30 s, and 68°C for 1 min, followed by 5 min at 68°C. Reactions were analyzed on a 7% Urea-PAGE gel and quantitated by Odyssey FC imager (LiCOR, Lincoln, NE) using the 700 nm wavelength channel. Recovery of open complex activity for mutant extracts was performed by pre-incubating the indicated extracts with 20 ng of recombinant Core Factor (rCF) for 20 min on ice with intermittent mixing prior to adding to the transcription reactions.

## Pol I initiation complex assembly

Oligonucleotides for assembling the promoter scaffold were purchased from Integrated DNA Technology (IDT, Coralville, IA). Sequences of the oligonucleotides used are: template strand, 5'-Biotin-ACTGGGGAATTCTTTCGAACTTGTCTTCAACTGCTTTCGCATGAAGTACCTCCCAACTACTTTTCCTCACACTTGTACTCCATGAC-3'; non-template strand, 5'-GTCATGGAGTACAAGTGTGAGGAAAAGTAGTTGGGTTTTTTTTTTTTTTTTTTGCAGTTGAAGACAAGTTCGAAAGAATTCCCCAGT-3'; RNA, 5'-AUGCGA-3'; upstream truncated non-template strand, 5'-TGAGGAAAAGTAGTTGGG-3'; downstream truncated non-template strand, 5'-GCAGTTGAAGACAAGTTCGAAAGAATTCCCCAGT-3'. Lyophilized oligos were first resuspended in ultra-pure water for a final concentration of 100 μM. Assembly of the template used for EM studies was done by mixing template strand, non-template strand and RNA oligos with a 1:1 molar ratio at a final concentration of 20 μM in ultra-pure water, and denaturing in boiling water bath for 5 min, followed by gradually cooling down to room temperature for 2 hr. Assembled nucleic acid templates were subsequently diluted to 2 μM concentration using ultra-pure water.

To assembly the Pol I Initial Transcribing Complex for cryo-EM analysis, 20 μl of purified yeast Pol I (~0.5 μM) was first incubated with 5-fold molar access of recombinant yeast Rrn3 protein at room temperature for 2 hr. Next, 2.5 μl of the biotinylated nucleic acid template (2 μM) was added, and incubated with Pol I/Rrn3 for 10 min at room temperature. Core Factor (~21.6 picomole) and TBP (~50 picomole) were then introduced into the mixture and incubated for another 10 min at room temperature. The protein-nucleic acid mixture was diluted two-fold with assembly buffer (12 mM HEPES pH 7.9, 0.12 mM EDTA, 12% glycerol, 8.25 mM MgCl$_2$, 60 mM KCl) plus 1 mM DTT, 2.5 ng/μl dI-dC, 5 μM ZnCl$_2$, and 0.05% NP-40 (Roche, Basel, Switzerland), and the incubation was extended

for another 10 min. Assembled complex was immobilized onto the magnetic streptavidin T1 beads (Invitrogen, Carlsbad, CA) which had been equilibrated with the assembly buffer. Following washing of the beads three times using a washing buffer (10 mM HEPES, 10 mM Tris, pH 7.9, 5% glycerol, 5 mM $MgCl_2$, 50 mM KCl, 1 mM DTT, 0.05% NP-40, 5 µM $ZnCl_2$), the complex was eluted by incubating the beads at room temperature for 1 hr with digestion buffer containing 10 mM HEPES, pH 7.9, 10 mM $MgCl_2$, 50 mM KCl, 1 mM DTT, 5% glycerol, 0.05% NP-40, 5 unit/µl EcoRI-HF (New England Biolabs). Samples used for negative stain EM were prepared the same way, but with only 10% of the material.

## Electron microscopy

Negative stain samples were prepared using 400 mesh copper grid containing a continuous carbon supporting layer. The grid was plasma cleaned for 10 s immediately before sample deposition using a Solarus plasma cleaner (Gatan, Pleasanton, CA) equipped with air at 25 W power. An aliquot (3 µl) of the purified sample was first briefly crosslinked using 0.05% glutaraldehyde on ice and under very low illumination conditions for 5 min, and then was placed onto the grid and allowed to absorb for 10 min at 100% humidity in a homemade humidity chamber kept under very low illumination conditions. It was subsequently stained by four successive 50 µl drops of 2% (w/v) uranyl formate solution, rocking 5 s, 10 s, 15 s and 20 s on the drops and followed by blotting till dryness. Data collection was performed using a JEOL 1400 transmission electron microscope operating at 120 kV at a nominal magnification of ×30,000 (3.71 Å per pixel). The data were collected using the Leginon data collection software (*Suloway et al., 2005*) on a Gatan 4k ×4 k CCD camera using low-dose procedures (20 e$^-$ Å$^{-2}$ exposures) and a range of defocus values (from −0.8 to −2.8 µm).

For preparing samples for cryo-EM analysis, the eluted complex was first briefly crosslinked using 0.185% glutaraldehyde on ice and under very low illumination conditions for 5 min. The sample (~4 µl) was then immediately loaded onto a 400 mesh Quantifoil grid containing 7 µm squares with 2 µm spacing (Quantifoil S 7/2, Electron Microscopy Sciences, Hatfield, PA). A thin carbon film was floated onto the grid before it was plasma cleaned for 10 s at 5 W power using a Solarus plasma cleaner (Gatan) equipped with air immediately before sample deposition. The sample was allowed to absorb to the grid for 30 min at 4°C and 100% humidity in a Vitrobot (FEI, Hillsboro, OR) under low illumination conditions, before blotted for 4 s at 25 force and plunge-frozen in liquid ethane. The frozen grids were stored in liquid nitrogen until imaging. Data collection was performed using a Titan Krios transmission electron microscope (FEI) operating at 300 kV. Data were acquired with a K2 Summit direct electron detector (Gatan) operating in super-resolution mode at a nominal magnification of 22,500 × (0.65 Å per pixel), using a range of defocus values (from −1.5 to −4.5 µm). 2351 movie series were collected using the MSI-Raster2 application of the Leginon data collection software (*Suloway et al., 2005*). 40-frame exposures were taken at 0.3 s per frame (12 s total exposure time), using a dose rate of 2 e$^-$ per pixel per second, corresponding to a total dose of 56.8 e$^-$ Å$^{-2}$ per movie series.

Data collection for the complex assembled on the truncated template (*Figure 3—figure supplement 2*) was performed using a JEOL 3200 microscope operating at 200 kV. Data were acquired with a K2 Summit direct electron detector (Gatan) operating in super-resolution mode at a nominal magnification of 15,000 × (1.18 Å per pixel), using a range of defocus values (from −2 to −5 µm). 176 movie series were collected using the MSI-Raster2 application of the Leginon data collection software (*Suloway et al., 2005*). 40-frame exposures were taken at 0.3 s per frame (12 s total exposure time), using a dose rate of 2 e$^-$ per pixel per second, corresponding to a total dose of 17.2 e$^-$ Å$^{-2}$ per movie series.

## Image processing and three-dimensional reconstruction

Negative stain data pre-processing was performed using the Appion processing environment (*Lander et al., 2009*). Particles were automatically selected from the micrographs using a difference of Gaussians (DoG) particle picker (*Voss et al., 2009*). The contract transfer function (CTF) of each micrograph was estimated using CTFFind3 (*Mindell and Grigorieff, 2003*), the phases were flipped using CTFFind3, and particle stacks were extracted using a box size of 108 × 108 pixels. A total of 45,964 particles were extracted for the Pol I Initial Transcribing Complex. Two-dimensional classification was conducted using iterative multivariate statistical analysis and multi-reference alignment

analysis (MSA-MRA) within the IMAGIC software (*van Heel et al., 1996*). Three-dimensional (3D) reconstruction of negative stained data was performed using an iterative multi-reference projection-matching approach containing libraries from the EMAN2 software package (*Tang et al., 2007*). The crystal structure of free Pol I (*Engel et al., 2013*; *Fernández-Tornero et al., 2013*) was low-pass filtered to 60 Å, which was used as the initial model for the reconstruction of the negatively stained Pol I Initial Transcribing Complex samples.

Cryo-EM data was pre-processed as follows. Movie frames were aligned using MotionCor2 (*Zheng et al., 2017*) to correct for specimen motion. The anisotropic magnification distortion [distortion angle, 28.2°; minor axis scale factor, 0.987; major axis scale factor, 1.013 (*Yu et al., 2016*) was corrected on the average of the aligned frames using mag_distortion_correct (*Grant and Grigorieff, 2015*). The corrected average micrographs were used for data pre-processing within the Appion processing environment (*Lander et al., 2009*). The CTF of each micrograph was estimated using Gctf (*Zhang, 2016*). Particles were automatically selected from the aligned micrographs using the DoG particle picker (*Voss et al., 2009*). A total of 1,135,584 and 283,672 particles were picked for the complex assembled on the full and the truncated templates, respectively. All three-dimensional (3D) classification and refinement steps were performed within RELION 2.0 (*Kimanius et al., 2016*).

For the Initial Transcribing Complex assembled on the full template (*Figure 1*), the initial set of 1,135,584 particles was subjected to an initial 3D auto-refinement, using the negative stain reconstruction of the complex low-pass filtered to 30 Å as the initial reference (*Figure 1—figure supplement 2C*). Subsequently, a 3D classification was performed on the picked particles using the 3D auto-refined model low-pass filtered to 30 Å as the initial reference (*Figure 1—figure supplement 2C*). Two out of five classes in this classification, corresponding to 71,591 and 77,222 particles, were indicative of well-preserved complex with sharp structural features and were selected for further processing. These two classes were combined and then subjected to 3D auto-refinement and a second round of 3D classification within RELION. One out of the five classes in the second round classification was only corresponded to Pol I by itself (*Figure 1—figure supplement 2C*). Therefore, this class was removed and the remaining particles were further refined, resulting in a reconstruction of the Pol I Initial Transcribing Complexat an overall resolution of 3.8 Å (*Figure 1—figure supplement 2C,D*). All resolutions reported herein correspond to the gold-standard Fourier shell correlation (FSC) using the 0.143 criterion (*Henderson et al., 2012*). Local resolution estimation indicated that the density for Core Factor was at lower resolution than Pol I, probably owing to conformational heterogeneity (*Figure 1—figure supplement 2C*). Subsequently, soft masks were applied around the Core Factor and Pol I density during further 3D refinement within RELION. This procedure resulted in an improved reconstruction of Core Factor and Pol I, with an overall resolution of 4.2 Å and 3.7 Å, respectively (*Figure 1—figure supplement 2C,D*). The reconstructions of classes 1–3 in the second round of classification showed distinct features. Therefore, they were separately refined, resulting in reconstructions at an overall resolution of 4.2 Å for State 1, 4.3 Å for State 2, and 3.9 Å for State 3 (*Figure 1—figure supplement 2C*).

For the Pre-Initiation Complex assembled on the truncated scaffold (*Figure 3—figure supplement 2A*), the 3D reconstruction was performed the same as above, except that only one round of 3D classification was conducted. This resulted in a reconstruction at an overall resolution of 6.9 Å after refinement (*Figure 3—figure supplement 2B*), using 43,843 particles.

The final density maps were automatically sharpened using the post-processing program within RELION and then filtered according to local resolution estimated within RELION 2.0. Volume segmentation, automatic rigid-body docking, figure and movie generation were performed using UCSF Chimera (*Goddard et al., 2007*; *Pettersen et al., 2004*).

Cryo-EM density maps have been deposited in the Electron Microscopy Data Bank (EMDB) under accession numbers EMD-8771 (full), EMD-8772 (Core Factor, local), EMD-8773 (Pol I, local), EMD-8774 (State 1), EMD-8775 (State 2), EMD-8776 (State 3), EMD-8777 (truncated construct). Model coordinates have been deposited in the Protein Data Bank (PDB) under accession numbers 5W5Y (full), 5W64 (State 1), 5W65 (State 2), 5W66 (State 3).

## Model building and computational protocols

To build the structure of the Pol I Initial Transcribing Complex, we used the known crystal structures of yeast Pol I (PDB ID: 4C3I) (*Fernández-Tornero et al., 2013*) as a starting point. To model downstream DNA, the DNA-RNA hybrid structure from a previous model of the yeast Pol I Elongation

Complex (PDB ID: 5M5X) (*Tafur et al., 2016*) was fit into the density and modified to include the DNA bubble region. To model upstream DNA, we used UCSF Chimera to generate B-form DNA and rigid body fit into the density. The upstream DNA register was determined by using the density of the Pol I/Core Factor complex with a truncated DNA scaffold (*Figure 3—figure supplement 2A*). The A49 tandem winged helix domain of Pol I (PDB ID: 3NFI) (*Geiger et al., 2010*) was rigid-body fit into the EM density map corresponding to the core of Pol I (State 3). The linker between A49 tandem winged helix and A49/A34.5 dimerization domain was built based on the EM density and secondary structure prediction (*Adamczak et al., 2004*; *Buchan et al., 2013*).

To model the yeast Rrn6 WD40 domain, the X-ray structure for the N-terminal domain of the human proto-oncogene Nup214 (PDB ID: 2OIT) (*Napetschnig et al., 2007*) was used as a template to construct the yeast WD40 structure. Residues 163–559, corresponding to the Rrn6 WD40, were aligned with Nup214/CAN residues 1–434 (PDB ID: 2OIT) (*Napetschnig et al., 2007*), initially with the PROMALS3D multiple sequences and structure alignment server (*Pei et al., 2007*), and then adjusted manually. We then employed density-guided homology modeling with RosettaCM to rebuild gaps in the sequence alignment and refine the entire structure. This approach relied on Rosetta's all-atom energy function augmented with an energy term to assess agreement to the experimental EM density (*Song et al., 2013*).

With no known structural homologues for the Rrn6 C-terminal helical bundle (HB) domain, Rrn7 (two cyclin folds) and Rrn11 (TPR domain with N- and C-terminal unstructured regions), we employed the GeneSilico protein structure prediction server (*Kozlowski and Bujnicki, 2012*) to predict the secondary structure and register the sequence (residues 570–777) in the density. We used Coot (*Emsley and Cowtan, 2004*) to construct the individual secondary-structure fragment to create a backbone only model by tracing the EM density. These secondary-structure fragments were then connected by manually extending the main-chain trace. The approximate orientations of residue side chains were built and manually inspected/corrected based on the electron density and secondary structure prediction. The N-terminal zinc ribbon domain of yeast TFIIB (PDB ID: 4BBR) (*Sainsbury et al., 2013*) was used as a template to build the N-terminal domain of Rrn7 and docked into a corresponding density region.

Subsequently, MDFF flexible fitting was applied to fit the initial model for core Pol to the density map while preserving the secondary structure (*Trabuco et al., 2008*). MDFF was followed up by the real-space refined against the respective maps using the Phenix package (*Adams et al., 2011*). The refined models for the core Pol I and the Core Factor subunits (Rrn6, Rrn7 and Rrn11) were separately rigid-body fitted into the three states of the complexes and the models combined to assemble the full Pol I Initial Transcribing Complex models. Molecular graphics and analyses were performed with the UCSF Chimera package developed by the Resource for Biocomputing, Visualization, and Informatics at the University of California, San Francisco (supported by NIGMS P41-GM103311) (*Pettersen et al., 2004*).

## Acknowledgements

We thank Dr. Jonathan Remis, Dr. Valorie Bowman, and Dr. Thomas Klose for assistance with microscope operation and data collection, Dr. Frank DiMaio and Brandon Frenz for assistance for initial model building, and Jason Pattie for computer support. We are grateful to Dr. Ishwar Radhakrishnan, Susan Fishbain, and Ryan Abdella for helpful discussion and comments on the manuscript. We also thank the staff at the Structural Biology Facility (SBF) of Northwestern University and Cryo-EM Facility of Purdue University for technical support. This work was supported by a Cornew Innovation Award from the Chemistry of Life Processes Institute at Northwestern University (to Y He), a Catalyst Award by the Chicago Biomedical Consortium with support from the Searle Funds at The Chicago Community Trust (to Y He), an Institutional Research Grant from the American Cancer Society (IRG-15-173-21 to Y He), SUNY Research Foundation (to BAK), the Central New York Community Foundation (to BAK), the US National Institutes of Health (NCI 5K22CA184235 to BAK, NIGMS GM110387 to II), the National Science Foundation (MCB-1149521 to II). Y Han is a recipient of the Chicago Biomedical Consortium Postdoctoral Research Grant. BAK is a Sinsheimer Scholar award from the Alexandrine and Alexander L Sinsheimer Fund. Computational resources were provided in part by XSEDE (CHE110042) and the National Energy Research for Scientific Computing Center (DE-AC02-05CH11231).

## Additional information

### Funding

| Funder | Grant reference number | Author |
| --- | --- | --- |
| Northwestern University | Cornew Innovation Award | Yuan He |
| Chicago Community Trust | Catalyst Award | Yuan He |
| American Cancer Society | Institutional Research Grant IRG-15-173-21 | Yuan He |
| SUNY Research Foundation | | Bruce A Knutson |
| Central New York Community Foundation | | Bruce A Knutson |
| National Cancer Institute | 5K22CA184235 | Bruce A Knutson |
| National Institute of General Medical Sciences | GM110387 | Ivaylo Ivanov |
| National Science Foundation | MCB-1149521 | Ivaylo Ivanov |
| Chicago Community Trust | Chicago Biomedical Consortium Postdoctoral Research Grant | Yan Han |
| Alexandrine and Alexander L. Sinsheimer Fund | Sinsheimer Scholar award | Bruce A Knutson |

The funders had no role in study design, data collection and interpretation, or the decision to submit the work for publication.

### Author contributions

YHa, Conceptualization, Data curation, Formal analysis, Methodology, Writing—original draft, Writing—review and editing; CY, THDN, AJJ, Data curation, Formal analysis, Methodology; II, BAK, Resources, Supervision, Writing—review and editing; YHe, Conceptualization, Resources, Formal analysis, Supervision, Funding acquisition, Investigation, Methodology, Writing—original draft, Writing—review and editing

### Author ORCIDs

Yan Han, http://orcid.org/0000-0002-1207-7756
Bruce A Knutson, http://orcid.org/0000-0003-3599-1302
Yuan He, http://orcid.org/0000-0002-1455-3963

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
