## [Decision Letter]

Thank you for submitting your article "Structure of RNA polymerase I pre-initiation complex reveals the mechanism of ATP-independent transcription initiation" for consideration by *eLife*. Your article has been reviewed by three peer reviewers, and the evaluation has been overseen by a Reviewing Editor and John Kuriyan as the Senior Editor. The following individuals involved in review of your submission have agreed to reveal their identity: Seth A Darst (Reviewer #2); Georgios Skiniotis (Reviewer #3).

The editors and the reviewers find that there is much of value in the paper, but have also raised some serious questions that will have to be addressed before your manuscript can be considered further. The reviewers have discussed the reviews with one another and the Reviewing Editor has drafted this decision to help you prepare a revised submission.

Summary:

This manuscript by He and colleagues reports the cryo-EM structure of yeast RNA polymerase I (Pol I) pre-initiation complex (PIC), which includes the Core Factor (CF) complex recognizing the upstream DNA promoter, at a nominal resolution of 3.8 Å. The authors also obtained a lower resolution cryo-EM reconstruction (6.9 Å) of PIC with a truncated scaffold DNA, enabling them to identify the register of the double stranded DNA in contact with CF. Further three-dimensional classification of the particle projections reveals three conformers (4.0-4.3 Å resolution) in the upstream DNA/CF region, which the authors rationalize as distinct functional states. Accordingly, the authors propose that the intrinsic mobility of CF plays a critical role in DNA melting, leading to transcription bubble opening. This model can explain why pol I, and perhaps pol III as well, do not require a TFIIH-like factor for this step. Overall the structures and potential insights are quite interesting. However, a number of important points must first be addressed regarding the precise phase of the transcription cycle that was trapped in the complexes visualized, as well as technical aspects of the structure determination.

Essential revisions:

1) The authors need to address the absence of Rrn3 from their structures and discuss the role of this initiation factor in binding to the open transcription bubble. The Introduction clearly outlines the requirement for Rrn3 during initiation, and the Materials and methods section states that this protein was added in 5-fold excess in the preparation of complexes for structural studies. However, there is no further mention of Rrn3 in the manuscript, with the exception of the Figure 6 legend. The reconstituted complex is active (Figure 1—figure supplement 1), so Rrn3 is presumably present at that point. If there is evidence that Rrn3 dissociates once the transcription bubble is opened, the authors should comment on that in the context of their structural analysis. A biochemical analysis of cross-linked samples, such as gel filtration/SDS PAGE or protein identification by mass spec., would address whether Rrn3 is present in complexes on the grid. The authors should also address the possibility that the subset of particles they used in reconstructions lacked Rrn3, where as some of the ~90% of particles that were excluded might contain Rrn3. Whatever the outcome, the role and fate of Rrn3 need to be explicitly addressed.

2) Given that the initiation factor, Rrn3, is missing and the polymerase complexes are formed on a bubble template with an RNA primer, it seems inaccurate to describe them as pre-initiation complexes (PICs). Without further justification, this raises questions about conclusions about the mechanisms of DNA unwinding. The Discussion and title should be revised to address more accurately the point in the transcription cycle captured by these structures.

3) The three conformations captured in the analysis represent a small subset of the different particles and conformations in the data set. The authors should provide evidence (either in the form of new experiments or citations of published observations, mutant studies) that support the functional relevance of these conformations.

4) Euler angle distributions of particle projections should be presented for each reconstruction.

[Editors' note: further revisions were requested prior to acceptance, as described below.]

Thank you for resubmitting your work entitled "Structural mechanism of ATP-independent transcription initiation by RNA polymerase I" for further consideration at *eLife*. Your revised article has been favorably evaluated by John Kuriyan (Senior editor), a Reviewing editor and one reviewer.

Your manuscript is now, in principle, acceptable for publication, but the editors require one small change to be made. In subsection “Molecular structure of CF” of the revised manuscript, the following statement is made:

"This highlights the strength of the cryo-EM approach in determining the structure of macromolecular assemblies as only a fraction of the material was required and yielded only slightly lower overall resolution (4.2 Å for cryo-EM versus 3.2 Å for crystallography)."

This paragraph appears to have been inserted during revision. The editors find it to be misleading, since (in an objective sense) 3.2 Å is much better than 4.2 Å, because the transition from one to the other involves a critical ability to resolve molecular features. The purpose of this new paragraph, it seems, is to highlight their paper with respect to the Cramer paper. The editors do not think that this is necessary, and in any case, disagree with this statement. Just in the next paragraph or two, the authors say "The resolution of our reconstruction of CF at 4.2 Å hinders us from confidently resolving the register of the upstream promoter sequence."

The paper can be accepted once this paragraph is removed. Note that this is not optional, but that no further editorial input is necessary, and editorial staff can make the decision to proceed.

The authors should also consider an issue that the editors wish to bring to their attention. The main text of the manuscript remains very difficult for the general reader (as opposed to the transcription specialist) to understand and follow. The principal reason for this is the excessive use of acronyms. Acronyms were particularly useful when journal articles were subject to severe space constraints, but *eLife* is not subject to such constraints. Why not say "Core Factor" everywhere, instead of CF? Why not say "General Transcription Factors" instead of GTF, or Initial Transcribing Complex instead of ITC? The latter acronym is particularly confusing. Connected to this issue is the fact that the labels associated with the structural figures are minimal. More, and clearer, labels would increase the impact of the figures. If these issues are improved, we believe the paper will be more accessible to the general reader, and possibly have more impact.

Editing the paper to address these issues of clarity is optional. The authors may wish to take a day or two to do this, or may have the PDF posted online and then edit the version of the paper that will be typeset for final publication. Either way, no further input from the editors is required. Please consult with editorial staff regarding this matter.

---

## [Author Response]

*Essential revisions:*

*1) The authors need to address the absence of Rrn3 from their structures and discuss the role of this initiation factor in binding to the open transcription bubble. The Introduction clearly outlines the requirement for Rrn3 during initiation, and the Materials and methods section states that this protein was added in 5-fold excess in the preparation of complexes for structural studies. However, there is no further mention of Rrn3 in the manuscript, with the exception of the Figure 6 legend. The reconstituted complex is active (Figure 1—figure supplement 1), so Rrn3 is presumably present at that point. If there is evidence that Rrn3 dissociates once the transcription bubble is opened, the authors should comment on that in the context of their structural analysis. A biochemical analysis of cross-linked samples, such as gel filtration/SDS PAGE or protein identification by mass spec., would address whether Rrn3 is present in complexes on the grid. The authors should also address the possibility that the subset of particles they used in reconstructions lacked Rrn3, where as some of the ~90% of particles that were excluded might contain Rrn3. Whatever the outcome, the role and fate of Rrn3 need to be explicitly addressed.*

We agree that Rrn3 plays an essential role during Pol I transcription initiation as stated in the Introduction section as well as the in vitro transcription assay shown in Figure 1—figure supplement 1. Although we included both Rrn3 and TBP in our assembly reactions, we did not observe densities in any of the classified reconstruction that could correspond to these two factors. As suggested by the reviewers, we repeated our reaction and monitored all fractions by SDS-PAGE and silver staining, which has been included as Figure 1—figure supplement 4. We observed on this gel that all of Rrn3 protein was in the unbound and the first wash fractions. In agreement with our structural studies, we did not observe a detectable band for Rrn3 on the gel in our elution fraction. However, we cannot rule out the possibility that Rrn3 was present at a substoichiometric level that is below the detection limit of silver staining. Alternatively, we cannot rule out the possibility that Rrn3 associates with Pol I in the unbound fraction that somehow failed to engage the nucleic acid template. Nevertheless, our data suggests that Rrn3 does not stably associate with the rest of Pol I initiation machinery under our experimental conditions, even though it is required for the activity of Pol I. Indeed, protocols involving incubation of Pol I over extended long period of time in the presence of excess recombinant Rrn3 have to be used in order to generate stable complex for structural studies (Blattner et al., 2011; Engel et al., 2016). Interestingly, we also observed a band that could result from TBP non-specifically binding to the DNA template (Figure 1—figure supplement 4).

As stated in the Introduction section, Rrn3 has been shown to disrupt the inactive dimeric form of Pol I in various reports (Blattner et al., 2011; Engel et al., 2016; Pilsl et al., 2016; Torreira et al., 2017). In addition, previous studies have also suggested that Rrn3 dissociates from Pol I after transcription initiation (Bier et al., 2004; Hirschler-Laszkiewicz et al., 2003; Milkereit and Tschochner, 1998). Paule and colleagues also reported that *Acanthamoeba castellani* rDNA promoter opening only requires TIF-IB (CF homolog) and Pol I (Kahl et al., 2000). Based on these published data and our observation, we speculate that Rrn3 functions at an earlier step during Pol I transcription initiation, where it stabilizes Pol I in an active monomeric form and helps recruiting Pol I to rDNA promoter. In a separate paper, Reeder and colleagues reported that Pol I can be recruited to the promoter in the absence of Rrn3, however this complex is inactive (Aprikian et al., 2001). This suggests that Rrn3 may also function post Pol I recruitment. This is also consistent with our structural study, as we included an RNA molecule in our open template, which may have resulted in bypassing the requirement for Rrn3 after the engagement of Pol I with the nucleic acid scaffold.

We have included a paragraph in the Results section describing all of the above.

*2) Given that the initiation factor, Rrn3, is missing and the polymerase complexes are formed on a bubble template with an RNA primer, it seems inaccurate to describe them as pre-initiation complexes (PICs). Without further justification, this raises questions about conclusions about the mechanisms of DNA unwinding. The Discussion and title should be revised to address more accurately the point in the transcription cycle captured by these structures.*

We agree that our usage of PIC may be confusing to the readers, especially the nucleic acid scaffold we used contains an artificial transcription bubble. In order to more accurately reflect the nature of our complex, we have revised our title to “Structural mechanism of ATP-independent transcription initiation by RNA polymerase I”, and changed the name of our complex to “initial transcribing complex (ITC)” throughout the manuscript where appropriate. We have also deleted the “CC model” and “OC model” for State 1 and State 2 in the descriptions of Figure 4 and its supplements.

We view the process of Pol I transcription initiation as a highly dynamic one. We believe that by using a bubble scaffold, we have bypassed the most dynamic process (from closed complex formation to bubble opening), while trapping the complex in a more stable ITC state. Nevertheless, the ITC structure still exhibits a high degree of flexibility (Figure 4 and Figure 4—figure supplement 1). In State 1, we observed that the upstream DNA trajectory was rotated the furthest away from the active site of Pol I. This also correlates with a wider cleft and the absence of the Rrn7 ZR domain or the A49 tWH domain (Figure 4). We speculate that in our State 1 structure, the position of CF and upstream DNA, and the conformation of Pol I might reflect those immediately after promoter DNA loading in a closed complex (CC). In line with this hypothesis, extending the upstream DNA using a B-type duplex DNA resulted in no clash to Pol I with a wide cleft (Figure 6—figure supplement 1). Therefore, we modeled the CC based on these features in our State 1 structure. To further justify this, we attempted the assembly of the complex on a closed duplex DNA scaffold, and were able to obtain a low resolution reconstruction from a negative staining dataset (see Figure 7). The overall architecture of the CC is similar to that of the ITC, even though the distinct views are more scarce and occupancy of CF is much lower. This is presumably due to the flexibility of the closed complex. Alternatively, because we cannot see DNA molecule in the negative stain reconstruction, we cannot rule out the possibility that the reconstructed complex is actually an open complex, similar to what has been observed for the yeast Pol II CC, in which the closed DNA template was melted spontaneously by engaging the initiation complex (Plaschka et al., 2016). Nevertheless, this new piece of data highlights the flexible nature of the Pol I initiation complex on a closed DNA template, and supports our modeling of the CC based on the State 1 structure.

Author response image 1.**DOI:**
http://dx.doi.org/10.7554/eLife.27414.023

In State 2, the upstream DNA trajectory was rotated closest to the active site of Pol I among the three states (Figure 4). Meanwhile, we also observed the stabilization of the Rrn7 ZR and linker regions. In addition, we performed an open complex assay, in which we demonstrated that Rrn7 ZR clearly functions in transcription bubble opening (see response to Comment #3 for details). This new data, together with our structural analysis, gives us more confidence that our State 2 structure truly mimics an open complex (OC). In State 3, we observed the stabilization of the A49 tWH domain and the gatekeeper helices spanning over the closed Pol I cleft (Figure 4). Given the role A49 tWH domain plays in transcription elongation (Beckouet et al., 2008; Geiger et al., 2010; Pilsl et al., 2016), as well as the observation that the active site cleft is in a closed state similar to that in the elongation complex, we believe that the State 3 structure reflects the bona fide initial transcribing state, prior to promoter escape.

*3) The three conformations captured in the analysis represent a small subset of the different particles and conformations in the data set. The authors should provide evidence (either in the form of new experiments or citations of published observations, mutant studies) that support the functional relevance of these conformations.*

We agree that additional experiments in support of our structural analysis will significantly improve this manuscript. To further test the hypothesis that the Rrn7 ZR domain potentially functions in promoter opening, we performed an open complex assay. We have included this data as Figure 4—figure supplement 3, updated the corresponding text in the Results section (subsection “Three functional states reveal conformational changes in Pol I ITC during transcription initiation “paragraph three), and inserted the experimental procedure in the Materials and methods section (subsection “Open complex assays”). Here, we will only briefly describe our finding. We used potassium permanganate to probe for the formation of transcription bubbles around the transcription start site. In one set of experiments, extract prepared from yeast strains bearing mutation in the Rrn7 N-terminus (deletion of the ZR domain) failed to form a transcription bubble. In another set of reactions, WT recombinant CF rescued the defect in the ZR deletion extract, whereas recombinant CF complexes with ZR deleted Rrn7 failed to do so. This data, together with previous finding that ZR deleted Rrn7 can still be recruited to rDNA promoter (Knutson et al., 2014), suggests that the Rrn7 N-terminal region promotes transcription bubble opening, in support of our State 2 as a OC mimic.

*4) Euler angle distributions of particle projections should be presented for each reconstruction.*

We have included the figures for the Euler angle distribution for each reconstruction in Figure 1—figure supplement 2 and Figure 3—figure supplement 2 as suggested.

[Editors' note: further revisions were requested prior to acceptance, as described below.]

*Your manuscript is now, in principle, acceptable for publication, but the editors require one small change to be made. In subsection “Molecular structure of CF” of the revised manuscript, the following statement is made:*

*"This highlights the strength of the cryo-EM approach in determining the structure of macromolecular assemblies as only a fraction of the material was required and yielded only slightly lower overall resolution (4.2 Å for cryo-EM versus 3.2 Å for crystallography)."*

*This paragraph appears to have been inserted during revision. The editors find it to be misleading, since (in an objective sense) 3.2 Å is much better than 4.2 Å, because the transition from one to the other involves a critical ability to resolve molecular features. The purpose of this new paragraph, it seems, is to highlight their paper with respect to the Cramer paper. The editors do not think that this is necessary, and in any case, disagree with this statement. Just in the next paragraph or two, the authors say "The resolution of our reconstruction of CF at 4.2 Å hinders us from confidently resolving the register of the upstream promoter sequence."*

*The paper can be accepted once this paragraph is removed. Note that this is not optional, but that no further editorial input is necessary, and editorial staff can make the decision to proceed.*

We agree with the reviewers and have deleted the following paragraph: "This highlights the strength of the cryo-EM approach in determining the structure of macromolecular assemblies as only a fraction of the material was required and yielded only slightly lower overall resolution (4.2 Å for cryo-EM versus 3.2 Å for crystallography)."

*The authors should also consider an issue that the editors wish to bring to their attention. The main text of the manuscript remains very difficult for the general reader (as opposed to the transcription specialist) to understand and follow. The principal reason for this is the excessive use of acronyms. Acronyms were particularly useful when journal articles were subject to severe space constraints, but eLife is not subject to such constraints. Why not say "Core Factor" everywhere, instead of CF? Why not say "General Transcription Factors" instead of GTF, or Initial Transcribing Complex instead of ITC? The latter acronym is particularly confusing. Connected to this issue is the fact that the labels associated with the structural figures are minimal. More, and clearer, labels would increase the impact of the figures. If these issues are improved, we believe the paper will be more accessible to the general reader, and possibly have more impact.*

We have reduced the use of acronyms throughout the manuscript, by replacing general transcription factor with GTF, pre-initiation complex with PIC, Core Factor with CF, initial transcribing complex with ITC.

We have made the labels in the following figures more accessible to the readers and uploaded new versions in this resubmission accordingly: Figure 2—figure supplement 1, Figure 2, Figure 3, Figure 4—figure supplement 1.

We have also included in the updated manuscript in subsection “Image processing and three-dimensional reconstruction” the final deposition accession numbers for our EM density maps and model coordinates in respective databases.